# Accelerated Regularized Learning in Finite $N$-Person Games

**Kyriakos Lotidis**
Stanford University
klotidis@stanford.edu

**Angeliki Giannou**
University of Wisconsin–Madison
giannou@wisc.edu

**Panayotis Mertikopoulos**
Univ. Grenoble Alpes, CNRS, Inria, Grenoble INP
LIG 38000 Grenoble, France
panayotis.mertikopoulos@imag.fr

**Nicholas Bambos**
Stanford University
bambos@stanford.edu

## Abstract

Motivated by the success of Nesterov's accelerated gradient algorithm for convex minimization problems, we examine whether it is possible to achieve similar performance gains in the context of online learning in games. To that end, we introduce a family of accelerated learning methods, which we call *"follow the accelerated leader"* (FTXL), and which incorporates the use of momentum within the general framework of regularized learning – and, in particular, the exponential / multiplicative weights algorithm and its variants. Drawing inspiration and techniques from the continuous-time analysis of Nesterov's algorithm, we show that FTXL converges locally to strict Nash equilibria at a *superlinear* rate, achieving in this way an exponential speed-up over vanilla regularized learning methods (which, by comparison, converge to strict equilibria at a *geometric*, linear rate). Importantly, FTXL maintains its superlinear convergence rate in a broad range of feedback structures, from deterministic, full information models to stochastic, realization-based ones, and even when run with bandit, payoff-based information, where players are only able to observe their individual realized payoffs.

## 1 Introduction

One of the most important milestones in convex optimization was Nesterov's accelerated gradient (NAG) algorithm, as proposed by Nesterov [38] in 1983. The groundbreaking achievement of Nesterov's algorithm was that it attained an $\mathcal{O}(1/T^2)$ rate of convergence in Lipschitz smooth convex minimization problems, thus bridging a decades-old gap between the $\mathcal{O}(1/T)$ convergence rate of ordinary gradient descent and the corresponding $\Omega(1/T^2)$ lower bound for said class [37]. In this way, Nesterov's accelerated gradient algorithm opened the door to acceleration in optimization, leading in turn to a wide range of other, likewise influential schemes – such as FISTA and its variants [3] – and jumpstarting a vigorous field of research that remains extremely active to this day.

Somewhat peculiarly, despite the great success that NAG has enjoyed in all fields where optimization plays a major role – and, in particular, machine learning and data science – its use has not percolated to the adjoining field of game theory as a suitable algorithm for learning Nash equilibria. Historically, the reasons for this are easy to explain: despite intense scrutiny by the community and an extensive corpus of literature dedicated to deconstructing the algorithm's guarantees, NAG's update structure remains quite opaque – and, to a certain extent, mysterious. Because of this, Nesterov's algorithm could not be considered as a plausible learning scheme that could be employed by boundedly rational *human* agents involved in a repeated game. Given that this was the predominant tenet in economic thought

38th Conference on Neural Information Processing Systems (NeurIPS 2024).

at the time, the use of Nesterov's algorithm in a game-theoretic context has not been extensively explored, to the best of our knowledge.

On the other hand, as far as applications to machine learning and artificial intelligence are concerned, the focus on *human* agents is no longer a limiting factor. In most current and emerging applications of game-theoretic learning – from multi-agent reinforcement learning to adversarial models in machine learning – the learning agents are algorithms whose computational capacity is only limited by the device on which they are deployed. In view of this, our paper seeks to answer the following question:

> *Can Nesterov's accelerated gradient scheme be deployed in a game-theoretic setting? And, if so, is it possible to achieve similar performance gains as in convex optimization?*

**Our contributions in the context of related work.** The answer to the above questions is not easy to guess. On the one hand, given that game theory and convex optimization are fundamentally different fields, a reasonable guess would be "no" – after all, finding a Nash equilibrium is a PPAD-complete problem [9], whereas convex minimization problems are solvable in polynomial time [7]. On the other, since in the context of online learning each player *would* have every incentive to use the most efficient unilateral optimization algorithm at their disposal, the use of NAG methods cannot be easily discarded from an algorithmic viewpoint.

Our paper examines if it is possible to obtain even a partially positive answer to the above question concerning the application of Nesterov's accelerated gradients techniques to learning in games. We focus throughout on the class of finite *N*-person games where, due to the individual concavity of the players' payoff functions, the convergence landscape of online learning in games is relatively well-understood – at least, compared to non-concave games. In particular, it is known that regularized learning algorithms – such as "follow the regularized leader" (FTRL) and its variants – converge locally to strict Nash equilibria at a geometric rate [18], and strict equilibria are the only locally stable and attracting limit points of regularized learning in the presence of randomness and/or uncertainty [11, 17, 23]. In this regard, we pose the question of (*i*) whether regularized learning schemes like FTRL can be accelerated; and (*ii*) whether the above properties are enhanced by this upgrade.

We answer both questions in the positive. First, we introduce an accelerated regularized scheme, in both continuous and discrete time, which we call *"follow the accelerated leader"* (FTXL). In continuous time, our scheme can be seen as a fusion of the continuous-time analogue of NAG proposed by Su, Boyd, and Candès [41] and the dynamics of regularized learning studied by Mertikopoulos & Sandholm [31] – see also [5, 6, 19, 24, 29, 32–36, 42] and references therein. We show that the resulting dynamics exhibit the same qualitative equilibrium convergence properties as the replicator dynamics of Taylor & Jonker [43] (the most widely studied instance of FTRL in continuous time). However, whereas the replicator dynamics converge to strict Nash equilibria at a linear rate, the FTXL dynamics converge *superlinearly*.

In discrete time, we likewise propose an algorithmic implementation of FTXL which can be applied in various information context: (*i*) *full information*, that is, when players observe their entire mixed payoff vector; (*ii*) *realization-based feedback*, i.e., when players get to learn the "what-if" payoff of actions that they did not choose; and (*iii*) *bandit, payoff-based feedback*, where players only observe their realized, in-game payoff, and must rely on statistical estimation techniques to reconstruct their payoff vectors. In all cases, we show that FTXL maintains the exponential speedup described above, and converges to strict Nash equilibria at a superlinear rate (though the subleading term in the algorithm's convergence rate becomes increasingly worse as less information is available). We find this feature of FTXL particularly intriguing as superlinear convergence rates are often associated to methods that are second-order in *space*, not *time*; the fact that this is achieved even with bandit feedback is quite surprising in this context.

Closest to our work is the continuous-time, second-order replicator equation studied by Laraki & Mertikopoulos [25] in the context of evolutionary game theory, and derived through a model of pairwise proportional imitation of "long-term success". The dynamics of [25] correspond to the undamped, continuous-time version of FTXL with entropic regularization, and the equilibrium convergence rate obtained by [25] agrees with our analysis. Other than that, the dynamics of Flåm & Morgan [12] also attempted to exploit a Newtonian structure, but they do not yield favorable convergence properties in a general setting. The inertial dynamics proposed in [26] likewise sought to leverage an inertial structure combined with the Hessian–Riemannian underpinnings of the replicator

dynamics, but the resulting replicator equation was not even well-posed (in the sense that its solutions exploded in finite time).

More recently, Gao & Pavel [15, 16] considered a second-order, inertial version of the dynamics of mirror descent in continuous games, and examined their convergence in the context of variational stability [34]. Albeit related at a high level to our work (given the link between mirror descent and regularized learning), the dynamics of Gao & Pavel [15, 16] are actually incomparable to our own, and there is no overlap in our techniques or results. Other than that, second-order dynamics in games have also been studied in continuous time within the context of control-theoretic passivity, yielding promising results in circumventing the impossibility results of Hart & Mas-Colell [21], cf. Gao & Pavel [13, 14], Mabrok & Shamma [30], Toonsi & Shamma [44], and references therein. However, the resulting dynamics are also different, and we do not see a way of obtaining comparable rates in our setting.

## 2 Preliminaries

In this section, we outline some notions and definitions required for our analysis. Specifically, we introduce the framework of finite $N$-player games, we discuss the solution concept of a Nash equilibrium, and we present the main ideas of regularized learning in games.

**2.1. Finite games.** In this work, we focus exclusively with finite games in normal form. Such games consist of a finite set of *players* $\mathcal{N} = \{1, \dots, N\}$, each of whom has a finite set of *actions* – or *pure strategies* – $\alpha_i \in \mathcal{A}_i$ and a *payoff function* $u_i \colon \mathcal{A} \to \mathbb{R}$, where $\mathcal{A} := \prod_{i \in \mathcal{N}} \mathcal{A}_i$ denotes the set of all possible action profiles $\alpha = (\alpha_1, \dots, \alpha_N)$. To keep track of all this, a finite game with the above primitives will be denoted as $\Gamma \equiv \Gamma(\mathcal{N}, \mathcal{A}, u)$.

In addition to pure strategies, players may also randomize their choices by employing *mixed strategies*, that is, by choosing probability distributions $x_i \in \mathcal{X}_i := \Delta(\mathcal{A}_i)$ over their pure strategies, where $\Delta(\mathcal{A}_i)$ denotes the probability simplex over $\mathcal{A}_i$. Now, given a *strategy profile* $x = (x_1, \dots, x_N) \in \mathcal{X} := \prod_{i \in \mathcal{N}} \mathcal{X}_i$, we will use the standard shorthand $x = (x_i; x_{-i})$ to highlight the mixed strategy $x_i$ of player $i$ against the mixed strategy profile $x_{-i} \in \mathcal{X}_{-i} := \prod_{j \neq i} \mathcal{X}_j$ of all other players. We also define:

1. The *mixed payoff* of player $i$ under $x$ as

$$u_i(x) = u_i(x_i; x_{-i}) = \sum_{\alpha_1 \in \mathcal{A}_1} \cdots \sum_{\alpha_N \in \mathcal{A}_N} x_{1\alpha_1} \dots x_{N\alpha_N} u_i(\alpha_1, \dots, \alpha_N) \qquad (1)$$

2. The *mixed payoff vector* of player $i$ under $x$ as

$$v_i(x) = \nabla_{x_i} u_i(x) = (u_i(\alpha_i; x_{-i}))_{\alpha_i \in \mathcal{A}_i} \qquad (2)$$

In words, $v_i(x)$ collects the expected rewards $v_{i\alpha_i}(x) := u_i(\alpha_i; x_{-i})$ of each action $\alpha_i \in \mathcal{A}_i$ of player $i \in \mathcal{N}$ against the mixed strategy profile $x_{-i}$ of all other players. Finally, we write $v(x) = (v_1(x), \dots, v_N(x))$ for the concatenation of the players' mixed payoff vectors.

In terms of solution concepts, we will say that $x^*$ is a *Nash equilibrium* (NE) if no player can benefit by unilaterally deviating from their strategy, that is

$$u_i(x^*) \geq u_i(x_i; x^*_{-i}) \quad \text{for all } x_i \in \mathcal{X}_i \text{ and all } i \in \mathcal{N}. \qquad \text{(NE)}$$

Moreover, we say that $x^*$ is a *strict Nash equilibrium* if (NE) holds as a strict inequality for all $x_i \neq x^*_i$, $i \in \mathcal{N}$, i.e., if any deviation from $x^*_i$ results in a strictly worse payoff for the deviating player $i \in \mathcal{N}$. It is straightforward to verify that a strict equilibrium $x^* \in \mathcal{X}$ is also *pure* in the sense that each player assigns positive probability only to a single pure strategy $\alpha^*_i \in \mathcal{A}_i$. Finally, we denote the *support* of a strategy $x$ as the set of actions with non-zero probability mass, i.e., $\text{supp}(x) = \{\alpha \in \mathcal{A} : x_\alpha > 0\}$.

**2.2. Regularized learning in games.** In the general context of finite games, the most widely used learning scheme is the family of algorithms and dynamics known as *"follow the regularized leader"* (FTRL). In a nutshell, the main idea behind FTRL is that each player $i \in \mathcal{N}$ plays a "regularized" best response to their cumulative payoff over time, leading to the continuous-time dynamics

$$\dot{y}_i(t) = v_i(x(t)) \qquad x_i(t) = Q_i(y_i(t)) \qquad \text{(FTRL-D)}$$

where

$$Q_i(y_i) = \arg\max_{x_i \in \mathcal{X}_i} \{\langle y_i, x_i \rangle - h_i(x_i)\} \tag{3}$$

denotes the *regularized best response* – or *mirror* – map of player $i \in \mathcal{N}$, and $h_i \colon \mathcal{X}_i \to \mathbb{R}$ is a strongly convex function known as the method's *regularizer*. Accordingly, in discrete time, this leads to the algorithm

$$y_{i,n+1} = y_{i,n} + \gamma \hat{v}_{i,n} \qquad x_{i,n} = Q_i(y_{i,n}) \tag{FTRL}$$

where $\gamma > 0$ is a hyperparameter known as the algorithm's *learning rate* (or *step-size*) and $\hat{v}_{i,n}$ is a black-box "payoff signal" that carries information about $v_i(x_n)$. In the simplest case, when players have full information about the game being played and the actions taken by their opponents, we have $\hat{v}_{i,n} = v_i(x_n)$; in more information-depleted environments (such as learning with payoff-based, bandit feedback), $\hat{v}_{i,n}$ is a reconstruction of $v_i(x_n)$ based on whatever information is at hand.

For concreteness, we close this section with the prototypical example of FTRL methods, the *exponential / multiplicative weights* (EW) algorithm. Going back to [2, 28, 45], this method is generated by the negentropy regularizer $h_i(x_i) = \sum_{\alpha_i \in \mathcal{A}_i} x_{i\alpha_i} \log x_{i\alpha_i}$, which yields the EW update rule

$$y_{i,n+1} = y_{i,n} + \gamma \hat{v}_{i,n} \qquad x_{i,n} = \Lambda_i(y_{i,n}) \coloneqq \frac{\exp(y_{i,n})}{\|\exp(y_{i,n})\|_1} \tag{EW}$$

and, in the continuous-time limit $\gamma \to 0$, the *exponential weights dynamics*

$$\dot{y}_i(t) = v_i(x(t)) \qquad x_i(t) = \Lambda_i(y_i(t)). \tag{EWD}$$

In the above, $\Lambda_i$ denotes the regularized best response induced by the method's entropic regularizer, which is known colloquially as a *logit best response* – or, even more simply, as the *logit map*. To make the notation more compact in the sequel, we will write $Q = (Q_i)_{i \in \mathcal{N}}$ and $\Lambda = (\Lambda_i)_{i \in \mathcal{N}}$ for the ensemble of the players' regularized / logit best response maps.

*Remark* 1. To streamline our presentation, in the main part of the paper, quantitative results will be stated for the special case of the EW setup above. In Appendix A, we discuss more general decomposable regularizers of the form $h_i(x_i) = \sum_{\alpha_i \in \mathcal{A}_i} \theta_i(x_i)$ where $\theta_i \colon [0,1] \to \mathbb{R}$ is continuous on $[0,1]$, and has $\theta''(x) > 0$ for all $x \in (0,1]$ and $\lim_{x \to 0^+} \theta'(x) = -\infty$. Although this set of assumptions can be relaxed, it leads to the clearest presentation of our results, so it will suffice for us.

*Remark* 2. Throughout the paper, we will interchangeably use $\dot{g}(t)$ and $dg/dt$ to denote the time derivative of $g(t)$. This dual notation allows us to adopt whichever form is most convenient in the given context. Moreover, for a process $g$, we will use the notation $g(t)$ for $t \geq 0$ if it evolves in continuous time, and $g_n$ for $n \in \mathbb{N}$ if it evolves in discrete time steps, omitting the time-index when it is clear from context.

## 3 Combining acceleration with regularization: First insights and results

In this section, we proceed to illustrate how Nesterov's accelerated gradient (NAG) method can be combined with FTRL. To keep things as simple as possible, we focus on the continuous-time limit, so we do not have to worry about the choice of hyperparameters, the construction of black-box models for the players' payoff vectors, etc.

**3.1. Nesterov's accelerated gradient algorithm.** We begin by discussing Nesterov's accelerated gradient algorithm as presented in Nesterov's seminal paper [38] in the context of unconstrained smooth convex minimization. Specifically, given a Lipschitz smooth convex function $f \colon \mathbb{R}^d \to \mathbb{R}$, the algorithm unfolds iteratively as

$$\begin{aligned} x_{n+1} &= w_n - \gamma \nabla f(w_n) \\ w_{n+1} &= x_{n+1} + \frac{n}{n+3}(x_{n+1} - x_n) \end{aligned} \tag{NAG}$$

where $w_1 = x_1$ is initialized arbitrarily and $\gamma > 0$ is a step-size parameter (typically chosen as $\gamma \leftarrow 1/L$ where $L$ is the Lipschitz smoothness modulus of $f$). The specific iterative structure of (NAG) – and, in particular the "3" in the denominator – can appear quite mysterious; nevertheless, (NAG) otherwise offers remarkable perfomance gains, improving in particular the rate of convergence

of gradient methods from $\mathcal{O}(1/T)$ to $\mathcal{O}(1/T^2)$ [38], and matching in this way the corresponding $\Omega(1/T^2)$ lower bound for the minimization of smooth convex functions [37].[1]

This groundbreaking result has since become the cornerstone of a vast and diverse literature expanding on the properties of (NAG) and trying to gain a deeper understanding of the "how" and "why" of its update structure. One perspective that has gained significant traction in this regard is the continuous-time approach of Su et al. [40, 41]; combining the two equations in (NAG) into

$$\frac{x_{n+1} - 2x_n + x_{n-1}}{\sqrt{\gamma}} = -\sqrt{\gamma}\,\nabla f(w_n) - \frac{3}{n+2}\frac{x_n - x_{n-1}}{\sqrt{\gamma}}, \tag{4}$$

they modeled (NAG) as a *heavy ball with vanishing friction* system of the form

$$\frac{d^2x}{dt^2} = -\nabla f(x) - \frac{3}{t}\frac{dx}{dt} \tag{HBVF}$$

The choice of terminology alludes to the fact that (HBVF) describes the dynamics of a heavy ball descending the landscape of $f$ under the potential field $F(x) = -\nabla f(x)$ with a vanishing kinetic friction coefficient (the $3/t$ factor in front of the momentum term $dx/dt$). In this interpretation, the mass of the ball accelerates the system, the friction term dissipates energy to enable convergence, and the vanishing friction coefficient quenches the impact of friction over time in order to avoid decelerating the system too much (so the system is, in a sense, "critically underdamped").

As was shown by Su et al. [41], an explicit Euler discretization of (HBVF) yields (NAG) with exactly the right momentum coefficient $n/(n+3)$; moreover, the rate of convergence of the continuous-time dynamics (HBVF) is the same as that of the discrete-time algorithm (NAG), and the energy function and Lyapunov analysis used to derive the former can also be used to derive the latter. For all these reasons, (HBVF) is universally considered as the *de facto* continuous-time analogue of (NAG), and we will treat it as such in the sequel.

**3.2. NAG meets FTRL.** To move from unconstrained convex minimization problems to finite $N$-person games – a constrained, non-convex, multi-agent, multi-objective setting – it will be more transparent to start with the continuous-time formulation (HBVF). Indeed, applying the logic behind (HBVF) to the (unconstrained) state variables $y$ of (FTRL-D), we obtain the *"follow the accelerated leader"* dynamics

$$\frac{d^2y}{dt^2} = v(Q(y)) - \frac{r}{t}\frac{dy}{dt} \tag{FTXL-D}$$

where the dynamics' driving force $F(y) = v(Q(y))$ is now given by the payoff field of the game, and the factor $r/t$, $r \geq 0$, plays again the role of a vanishing friction coefficient. To avoid confusion, we highlight that in the case of regularized learning, the algorithm's variable that determines the evolution of the system in an autonomous way is the "score variable" $y$, not the "strategy variable" $x$ (which is an ancillary variable obtained from $y$ via the regularized choice map $Q$).

In contrast to (EWD), the accelerated dynamics (FTXL-D) are second-order in time, a fact with fundamental ramifications, not only from a conceptual, but also from an operational viewpoint. Focusing on the latter, we first note that (FTXL-D) requires two sets of initial conditions, $y(0)$ and $\dot{y}(0)$, the latter having no analogue in the first-order setting of (FTRL-D). In general, the evolution of the system depends on both $y(0)$ and $\dot{y}(0)$, but since this would introduce an artificial bias toward a certain direction, we will take $\dot{y}(0) = 0$, in tune with standard practice for (NAG) [41].

We also note that (FTXL-D) can be mapped to an equivalent autonomous first-order system with double the variables: specifically, letting $p = \dot{y}$ denote the players' (*payoff*) *momentum*, (FTXL-D) can be rewritten as

$$\frac{dy}{dt} = p \qquad \frac{dp}{dt} = v(Q(y)) - \frac{r}{t}p \tag{5}$$

with $y(0)$ initialized arbitrarily and $p(0) = \dot{y}(0)$. In turn, (5) yields $p(t) = t^{-r}\int_0^t \tau^r v(Q(y(\tau)))\,d\tau$, so $p(t)$ can be seen as a weighted aggregate of the players' payoffs up to time $t$: if $r = 0$ (the undamped regime), all information enters $p(t)$ with the same weight; if $r > 0$, past information is discounted relative to more recent observations; and, in the overdamped limit $r \to \infty$, all weight is assigned to the current point in time, emulating in this way the first-order system (FTRL-D).

---

[1]There are, of course, many other approaches to acceleration, that we cannot cover here; for a discussion of the popular "linear coupling" approach of Allen-Zhu & Orecchia [1], see Appendix F.

**3.3. First insights and results.** From an operational standpoint, the main question of interest is to specify the equilibrium convergence properties of (FTXL-D) – and, later in the paper, its discrete-time analogue. To establish a baseline, the principal equilibrium properties of its first-order counterpart can be summarized as follows: (*i*) strict Nash equilibria are locally stable and attracting under (FTRL-D) [23, 31];[2] (*ii*) the dynamics do not admit any other such points (that is, stable and attracting) [11]; and (*iii*) quantitively, in the case of (EWD), the dynamics converge locally to strict Nash equilibria at a *geometric* rate of the form $\|x(t) - x^*\| = \mathcal{O}(\exp(-ct))$ for some $c > 0$ [31].

Our first result below shows that the accelerated dynamics (FTXL-D) exhibit an exponential speed-up relative to (FTRL-D), and the players' orbits converge to strict Nash equilibria at a *superlinear* rate:

**Theorem 1.** *Let $x^*$ be a strict Nash equilibrium of $\Gamma$, and let $x(t) = Q(y(t))$ be a solution orbit of* (FTXL-D). *If $x(0)$ is sufficiently close to $x^*$, then $x(t)$ converges to $x^*$; in particular, if* (FTXL-D) *is run with logit best responses (that is, $Q \leftarrow \Lambda$), we have*

$$\|x(t) - x^*\|_\infty \leq \exp\left(C - \frac{ct^2}{2(r+1)}\right) \tag{6}$$

*where $C > 0$ is a constant that depends only on the initialization of* (FTXL-D) *and*

$$c = \frac{1}{2} \min_{i \in \mathcal{N}} \min_{\beta_i \notin \mathrm{supp}(x_i^*)} [u_i(x_i^*; x_{-i}^*) - u_i(\beta_i; x_{-i}^*)] > 0 \tag{7}$$

*is the minimum payoff difference at equilibrium.*

Theorem 1 (which we prove in Appendix B) is representative of the analysis to come, so some remarks are in order. First, we should note that the explicit rate estimate (6) is derived for the special case of logit best responses, which underlie all exponential / multiplicative weights algorithms. To the best of our knowledge, the only comparable result in the literature is the similar rate provided in [25] for the case $r = 0$. In the case of a general regularizer, an analogous speed-up is observed, but the exact expressions are more involved, so we defer them to Appendix B. A second important point concerns whether the rate estimate (6) is tight or not. Finally, the neighborhood of initial conditions around $x^*$ is determined by the minimum payoff difference at equilibrium and is roughly $\mathcal{O}(c)$ in diameter; we defer the relevant details of this discussion to Appendix B.

To answer this question – and, at the same time get a glimpse of the proof strategy for Theorem 1 – it will be instructive to consider a single-player game with two actions. Albeit simple, this toy example is not simplistic, as it provides an incisive look into the problem, and will be used to motivate our design choices in the sequel.

**Example 3.1.** Consider a single-player game $\Gamma$ with actions A and B such that $u(A) - u(B) = 1$, so the (dominant) strategy $x^* = (1, 0)$ is a strict Nash equilibrium. Then, letting $z = y_A - y_B$, (FTXL-D) readily yields

$$\frac{d^2z}{dt^2} = \frac{d^2y_A}{dt^2} - \frac{d^2y_B}{dt^2} = u(A) - u(B) - \frac{r}{t}\left[\frac{dy_A}{dt} - \frac{dy_B}{dt}\right] = 1 - \frac{r}{t}\frac{dz}{dt}. \tag{8}$$

As we show in Appendix B, this non-autonomous differential equation can be solved exactly to yield $z(t) = z(0) + t^2/[2(r+1)]$, and hence

$$\|x(t) - x^*\|_\infty = \frac{1}{1 + \exp(z(t))} \sim \exp\left(-z(0) - \frac{t^2}{2(r+1)}\right). \tag{9}$$

Since $c = u(A) - u(B) = 1$, the rate (9) coincides with that of Theorem 1 up to a factor of $1/2$. This factor is an artifact of the analysis and, in fact, it can be tightened to $(1 - \varepsilon)$ for arbitrarily small $\varepsilon > 0$; we did not provide this more precise expression to lighten notation. By contrast, the factor $2(r+1)$ in (6) *cannot* be lifted; this has important ramifications which we discuss below. ◆

The first conclusion that can be drawn from Example 3.1 is that the rate estimate of Theorem 1 is tight and cannot be improved in general. In addition, and in stark contrast to (NAG), Example 3.1

---

[2]Recall here that $x^* \in \mathcal{X}$ is said to be (*i*) *Lyapunov stable* (or simply *stable*) if every orbit $x(t)$ of the dynamics that starts close enough to $x^*$ remains close enough to $x^*$ for all $t \geq 0$; (*ii*) *attracting* if $\lim_{t\to\infty} x(t) = x^*$ for every orbit $x(t)$ that starts close enough to $x^*$; and (*iii*) *asymptotically stable* if it is both stable and attracting. For an introduction to the theory of dynamical systems, cf. Shub [39] and Hirsch et al. [22].

shows that the optimal value for the friction parameter is $r = 0$ (at least from a min-max viewpoint, as this value yields the best possible lower bound for the rate). Of course, this raises the question as to whether this is due to the continuous-time character of the policy;[3] however, as we show in detail in Appendix C, this is *not* the case: the direct handover of (NAG) to Example 3.1 yields the exact same rate (though the proof relies on a significantly more opaque generating function calculation).

In view of all this, it becomes apparent that friction only *hinders* the equilibrium convergence properties of accelerated FTRL schemes in our game-theoretic setting. On that account, we will continue our analysis in the undamped regime $r = 0$.

## 4   Accelerated learning: Analysis and results

**4.1. The algorithm.**   To obtain a bona fide, algorithmic implementation of the continuous-time dynamics (FTXL-D), we will proceed with the same explicit, finite-difference scheme leading to the discrete-time algorithm (NAG) from the continuous-time dynamics (HBVF) of Su et al. [41]. Specifically, taking a discretization step $\gamma > 0$ in (FTXL-D) and setting the scheme's friction parameter $r$ to zero (which, as we discussed at length in the previous section, is the optimal choice in our setting), a straightforward derivation yields the basic update rule

$$[y_{i,n+1} - 2y_{i,n} + y_{i,n-1}]/\gamma^2 = \hat{v}_{i,n} \quad \text{for all } i \in \mathcal{N} \text{ and all } n = 1, 2, \ldots \tag{10}$$

In the above, just as in the case of (FTRL), $\hat{v}_{i,n} \in \mathbb{R}^{\mathcal{A}_i}$ denotes a black-box "payoff signal" that carries information about the mixed payoff vector $v_i(x_n)$ of player $i$ at the current strategy profile $x_n$ (we provide more details on this below).

Alternatively, to obtain an equivalent first-order iterative rule (which is easier to handle and discuss), it will be convenient to introduce the momentum variables $p_n = (y_n - y_{n-1})/\gamma$. Doing just that, a simple rearrangement of (10) yields the *"follow the accelerated leader"* scheme

$$y_{i,n+1} = y_{i,n} + \gamma p_{i,n+1} \qquad p_{i,n+1} = p_{i,n} + \gamma \hat{v}_{i,n} \qquad x_{i,n} = Q_i(y_{i,n}) \,. \tag{FTXL}$$

The algorithm (FTXL) will be our main object of study in the sequel, and we will examine its convergence properties under three differerent models for $\hat{v}_n$:

1. *Full information*, i.e., players get to access their full, mixed payoff vectors:

$$\hat{v}_{i,n} = v_i(x_n) \qquad \text{for all } i \in \mathcal{N}, n = 1, 2, \ldots \tag{11a}$$

2. *Realization-based feedback*, i.e., after choosing an action profile $\alpha_n \sim x_n$, each player $i \in \mathcal{N}$ observes (or otherwise calculates) the vector of their counterfactual, "what-if" rewards, namely

$$\hat{v}_{i,n} = v_i(\alpha_n) \qquad \text{for all } i \in \mathcal{N}, n = 1, 2, \ldots \tag{11b}$$

3. *Bandit / Payoff-based feedback*, i.e., each player only observes their current reward, and must rely on statistical estimation techniques to reconstruct an estimate of $v_i(x_n)$. For concreteness, we will consider the case where players employ a version of the so-called *importance-weighted estimator*

$$\hat{v}_{i,n} = \text{IWE}(x_{i,n}; \alpha_{i,n}) \qquad \text{for all } i \in \mathcal{N}, n = 1, 2, \ldots \tag{11c}$$

which we describe in detail later in this section.

Of course, this list of information models is not exhaustive, but it is a faithful representation of most scenarios that arise in practice, so it will suffice for our purposes.

Now before moving forward with the analysis, it will be useful to keep some high-level remarks in mind. The first is that (FTXL) shares many similarities with (FTRL), but also several notable differences. At the most basic level, (FTRL) and (FTXL) are both "stimulus-response" schemes in the spirit of Erev & Roth [10], that is, players "respond" with a strategy $x_{i,n} = Q_i(y_{i,n})$ to a "stimulus" $y_{i,n}$ generated by the observed payoff signals $\hat{v}_{i,n}$. In this regard, both methods adhere to the online learning setting (and, in particular, to the regularized learning paradigm).

---

[3]The reader might also wonder if the use of a *non-vanishing* friction coefficient $- r\dot{y}$ instead of $(r/t)\dot{y} -$ could be beneficial to the convergence rate of (FTXL-D). As we show in Appendices B and C, this leads to significantly worse convergence rates of the form $\|x(t) - x^*\|_\infty \sim \exp(-\Theta(t))$ for all $r > 0$.

However, unlike (FTRL), where players respond to the aggregate of their payoff signals – the process $y_n$ in (FTRL) – the accelerated algorithm (FTXL) introduces an additional aggregation layer, which expresses how players "build momentum" based on the same payoff signals – the process $p_n$ in (FTXL). Intuitively, we can think of these two processes as the "position" and "momentum" variables of a classical inertial system, not unlike the heavy-ball dynamics of Su et al. [41]. The only conceptual difference is that, instead of rolling along the landscape of a (convex) function, the players now track the "mirrored" payoff field $\hat{v}(y) := v(Q(y))$.

In the rest of this section, we proceed to examine in detail the equilibrium convergence properties of (FTXL) under each of the three models detailed in Eqs. (11a)–(11c) in order.

**4.2. Accelerated learning with full information.** We begin with the full information model (11a). This is the most straightforward model (due to the absence of randomness and uncertainty) but, admittedly, also the least realistic one. Nevertheless, it will serve as a useful benchmark for the rest, and it will allow us to introduce several important notions.

Before we state our result, it is important to note that a finite game can have multiple strict Nash equilibria, so global convergence results are, in general, unattainable; for this reason, we analyze the algorithm's local convergence landscape. In this regard, Theorem 2 below shows that (FTXL) with full information achieves a *superlinear* local convergence rate to strict Nash equilibria:

**Theorem 2.** *Let $x^*$ be a strict Nash equilibrium of $\Gamma$, and let $x_n = Q(y_n)$ be the sequence of play generated by (FTXL) with full information feedback of the form (11a). If $x_1$ is initialized sufficiently close to $x^*$, then $x_n$ converges to $x^*$; in particular, if (FTXL) is run with logit best responses (that is, $Q \leftarrow \Lambda$), we have*

$$\|x_T - x^*\|_\infty \leq \exp\left(C - c\gamma^2 \frac{T(T-1)}{2}\right) = \exp(-\Theta(T^2)) \tag{12}$$

*where $C > 0$ is a constant that depends only on the initialization of (FTXL) and*

$$c = \frac{1}{2} \min_{i \in \mathcal{N}} \min_{\beta_i \notin \mathrm{supp}(x_i^*)} [u_i(x_i^*; x_{-i}^*) - u_i(\beta_i; x_{-i}^*)] > 0 \tag{13}$$

*is the minimum payoff difference at equilibrium.*

To maintain the flow of our discussion, we defer the proof of Theorem 2 to Appendix C. Instead, we only note here that, just as in the case of (HBVF) and (NAG), Theorem 2 provides essentially the same rate of convergence as its continuous-time counterpart, Theorem 1, modulo a subleading term which has an exponentially small impact on the rate of convergence. In particular, we should stress that the *superlinear* convergence rate of (FTXL) exhibits an exponential speedup relative to (FTRL), which is known to converge at a geometric rate $\|x_T - x^*\|_\infty = \exp(-\Theta(T))$. This is in direct correspondence to what we observe in continuous time, showing in particular that the continuous-time dynamics (FTXL-D) are a faithful representation of (FTXL).

We should also stress here that superlinear convergence rates are typically associated with methods that are second-order *in space*, in the sense that they employ Hessian-like information – like Newton's algorithm – not second-order *in time* – like (NAG) and (FTXL). We find this observation particularly intriguing as it suggests that accelerated rates can be observed in the context of learning in games without having to pay the excessively high compute cost of second-order methods in optimization.

**4.3. Accelerated learning with realization-based feedback.** We now turn to the realization-based model (11b), where players can only assess the rewards of their pure actions in response to the *realized* actions of all other players. In words, $\hat{v}_{i,n} = v_i(\alpha_n)$ collects the payoffs that player $i \in \mathcal{N}$ would have obtained by playing each of their pure actions $\alpha_i \in \mathcal{A}_i$ against the action profile $\alpha_{-i,n}$ adopted by the rest of the players.

In contrast to the full information model (11a), the realization-based model is stochastic in nature, so our convergence results will likewise be stochastic. Nevertheless, despite the added layer of uncertainty, we show that (FTXL) with realization-based feedback maintains a superlinear convergence rate with high probability:

**Theorem 3.** *Let $x^*$ be a strict Nash equilibrium of $\Gamma$, fix some confidence level $\delta > 0$, and let $x_n = Q(y_n)$ be the sequence of play generated by (FTXL) with realization-based feedback as per*

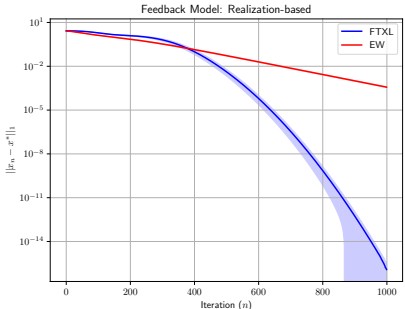
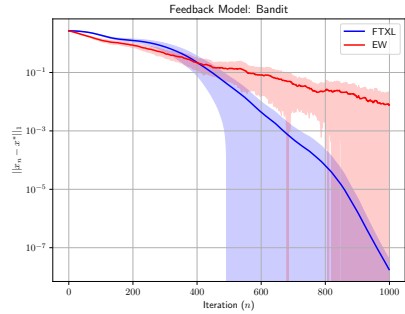

**(a)** Zero-sum game: Realization-based feedback  **(b)** Zero-sum game: Bandit feedback

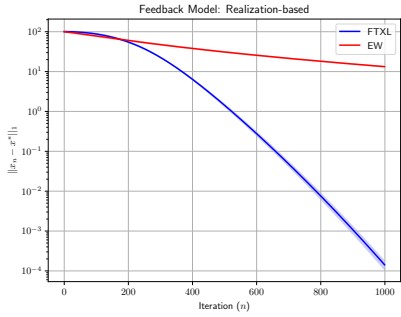
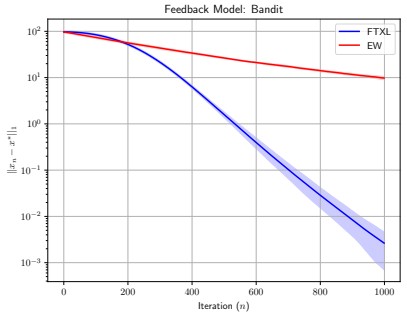

**(c)** Congestion game: Realization-based feedback  **(d)** Congestion game: Bandit feedback

**Figure 1:** Performance evaluation of (FTXL) in a zero-sum and a congestion game under realization-based and bandit feedback. Solid lines represent average values, while shaded regions enclose ±1 standard deviation. The plots are in logarithmic scale.

(11b) *and a sufficiently small step-size $\gamma > 0$. Then there exists a neighborhood $\mathcal{U}$ of $x^*$ such that*

$$\mathbb{P}(x_n \to x^* \text{ as } n \to \infty) \geq 1 - \delta \qquad \text{if } x_1 \in \mathcal{U}. \tag{14}$$

*In particular, if* (FTXL) *is run with logit best responses (that is, $Q \leftarrow \Lambda$), there exist positive constants $C, c > 0$ as in Theorem 2 such that on the event $\{x_n \to x^* \text{ as } n \to \infty\}$:*

$$\|x_T - x^*\|_\infty \leq \exp\left(C - c\gamma^2 \frac{T(T-1)}{2} + \frac{3}{5}c\gamma^{5/3}T^{5/3}\right) = \exp\left(-\Theta(T^2)\right). \tag{15}$$

What is particularly surprising in Theorem 3 is that, (FTXL) maintains the accelerated superlinear rate of Theorem 2 – and, likewise, the exponential speedup relative to (FTRL) – *despite* the randomness and uncertainty involved in the realization-based model (11b). The salient point enabling this feature of (FTXL) is that $\hat{v}_n$ can be expressed as

$$\hat{v}_n = v(x_n) + U_n \tag{16}$$

where $U_n \in \prod_i \mathbb{R}^{\mathcal{A}_i}$ is an almost surely bounded conditionally zero-mean stochastic perturbation, that is, $\mathbb{E}[U_n \mid \mathcal{F}_n] = 0$, where $\mathcal{F}_n := \sigma(x_1, \ldots, x_n)$ denotes the history of play up to (and including) time $n$. Thanks to the boundedness of (16), we are able to derive a series of probabilistic estimates showing that, with high probability (and, in particular, greater than $1 - \delta$), the contribution of the noise in the algorithm's rate becomes subleading, thus allowing the superlinear rate of Theorem 2 to emerge. As in the case of Theorem 2, we defer the proof of Theorem 3 to the appendix.

**4.4. Bandit feedback.** The last framework we consider is the bandit model where players only observe their realized rewards, a scalar from which they must reconstruct their entire payoff vector. To do so, a standard technique from the multi-armed bandit literature is the so-called *importance weighted estimator* (IWE) [8, 27], defined in our setting as

$$\hat{v}_{i\alpha_i,n} = \frac{\mathbb{1}\{\alpha_{i,n} = \alpha_i\}}{\hat{x}_{i\alpha_{i,n}}} u_i(\alpha_i; \alpha_{-i,n}) \tag{IWE}$$

where $\hat{x}_{i,n} = (1 - \varepsilon_n)x_{i,n} + \varepsilon_n \operatorname{unif}_{\mathcal{A}_i}$ is a mixture of $x_{i,n}$ and the uniform distribution on $\mathcal{A}_i$ (a mechanism known in the literature as *explicit exploration*). Importantly, this estimator is unbiased relative to the perturbed strategy $\hat{x}_{x_n}$, which thus incurs an $\mathcal{O}(\varepsilon_n)$ non-zero-sum error to the estimation of $v_i(x_n)$. This error can be made arbitrarily small by taking $\varepsilon_n \to 0$ but, in doing so, the variance of $\hat{v}_{i,n}$ diverges, leading to a bias-variance trade-off that is difficult to tame.

Despite these added difficulties, we show below that (FTXL) maintains its superlinear convergence rate even with bandit, payoff-based feedback:

**Theorem 4.** *Let $x^*$ be a strict Nash equilibrium of $\Gamma$, fix some confidence level $\delta > 0$, and let $x_n = Q(y_n)$ be the sequence of play generated by* (FTXL) *with bandit feedback of the form* (11c), *an IWE exploration parameter $\varepsilon_n \propto 1/n^{\ell_\varepsilon}$ for some $\ell_\varepsilon \in (0, 1/2)$, and a sufficiently small step-size $\gamma > 0$. Then there exists a neighborhood $\mathcal{U}$ of $x^*$ in $\mathcal{X}$ such that*

$$\mathbb{P}(x_n \to x^* \text{ as } n \to \infty) \geq 1 - \delta \qquad \text{if } x_1 \in \mathcal{U}. \tag{17}$$

*In particular, if* (FTXL) *is run with logit best responses (that is, $Q \leftarrow \Lambda$), there exist positive constants $C, c > 0$ as in Theorem 2 such that on the event $\{x_n \to x^* \text{ as } n \to \infty\}$*

$$\|x_T - x^*\|_\infty \leq \exp\left(C - c\gamma^2 \frac{T(T-1)}{2} + \frac{5}{9}c\gamma^{9/5}T^{9/5}\right) = \exp\left(-\Theta(T^2)\right). \tag{18}$$

Theorem 4 (which we prove in Appendix D) shows that, despite the degradation of the subleading term, (FTXL) retains its superlinear convergence rate even with bandit, payoff-based feedback (for a numerical demonstration, see Fig. 1 above). We find this feature of (FTXL) particularly important as it shows that the algorithm remains exceptionally robust in the face of randomness and uncertainty, even as we move toward increasingly information-starved environments – from full information, to realization-based observations and, ultimately, to bandit feedback. This has important ramifications from an operational standpoint, which we intend to examine further in future work.

**4.5. Numerical Experiments.** We conclude this section with a series of numerical simulations to validate the performance of (FTXL). To this end, we consider two game paradigms, (i) a 2-player zero-sum game, and (ii) a congestion game.

**Zero-sum Games.** First, we consider a 2-player zero-sum game with actions $\{\alpha_1, \alpha_2, \alpha_3\}$ and $\{\beta_1, \beta_2, \beta_3\}$, and payoff matrix

$$P = \begin{pmatrix} (2, -2) & (1, -1) & (2, -2) \\ (-2, 2) & (-1, 1) & (-2, 2) \\ (-2, 2) & (-1, 1) & (-2, 2) \end{pmatrix}$$

Here, the rows of $P$ correspond to the actions of player $A$ and the columns to the actions of player $B$, while the first item of each entry of $P$ corresponds to the payoff of $A$, and the second one to the payoff of $B$. Clearly, the action profile $(\alpha_1, \beta_2)$ is a strict Nash equilibrium.

**Congestion Games.** As a second example, we consider a congestion game with $N = 100$ and 2 roads, $r_1$ and $r_2$, with costs $c_1 = 1.1$ and $c_2 = d/N$ where $d$ is the number of drivers on $r_2$. In words, $r_1$ has a fixed delay equal to 1.1, while $r_2$ has a delay proportional to the drivers using it. Note, that the strategy profile where all players are using $r_2$ is a strict Nash equilibrium.

In Fig. 1, we assess the convergence of (FTXL) with logit best responses, under realization-based and bandit feedback, and compare it to the standard (EW) with the same level of information. The figures verify that (FTXL) outperforms (EW) regarding the convergence to a strict Nash equilibrium both for the realization-based and the bandit feedback, as expected from the theoretical findings. Specifically, they validate the faster convergence rate of (FTXL) compared to that of the (EW) algorithm. Moreover, we observe that both algorithms perform worse under bandit feedback than under realization-based feedback. This behavior is expected as less information becomes available. More details can be found in Appendix E.

## Acknowledgments and Disclosure of Funding

This research was supported in part by the French National Research Agency (ANR) in the framework of the PEPR IA FOUNDRY project (ANR-23-PEIA-0003), the "Investissements d'avenir program" (ANR-15-IDEX-02), the LabEx PERSYVAL (ANR-11-LABX-0025-01), MIAI@Grenoble Alpes (ANR-19-P3IA-0003), the project IRGA2024-SPICE-G7H-IRG24E90. PM is also with the Archimedes Research Unit – Athena RC – Department of Mathematics, National & Kapodistrian University of Athens, and his research was partially funded by project MIS 5154714 of the National Recovery and Resilience Plan Greece 2.0 funded by the European Union under the NextGenerationEU Program.

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

# Appendix

In [Appendix A](#), we discuss how our findings can be extended to general regularizers. Subsequently, [Appendices B](#) and [C](#) contain the technical proofs for the continuous and discrete time algorithms, respectively. Following this, [Appendix D](#) provides the convergence results of [(FTXL)](#) under partial information, specifically under realization-based and bandit feedback. We conclude this section with [Appendix E](#), which presents the details of the numerical experiments.

## A  Auxiliary results for general regularizers

In this appendix, we briefly discuss how to obtain the convergence of [(FTXL)](#) for mirror maps $Q$ *beyond* the logit map $\Lambda$. Namely, we consider regularizers that are decomposable, i.e., $h_i(x_i) = \sum_{\alpha_i \in \mathcal{A}_i} \theta_i(x_{\alpha_i})$ such that $\theta_i \colon [0,1] \to \mathbb{R}$ is continuous on $[0,1]$, twice differentiable on $(0,1]$ and strongly convex with $\theta_i'(0^+) = -\infty$.

**Lemma A.1.** *Suppose that $x_n = Q(y_n)$ and for all $\alpha \in \mathcal{A}, \alpha \neq \alpha^*$, it holds that $y_{\alpha,n} - y_{\alpha^*,n} \to -\infty$ as $n \to \infty$. Then, $x_n$ converges to $x^*$, where $x^*$ is a point mass at $\alpha^*$. Moreover, it holds that:*

$$\|x_n - x^*\|_\infty \leq \sum_{\alpha \neq \alpha^*} (\theta')^{-1}\big(\theta'(1) + y_{\alpha,n} - y_{\alpha^*,n}\big) \tag{A.1}$$

*Proof.* First, note that for $x = Q(y)$, we have that $x$ is the solution of the following optimization problem

$$Q(y) = \arg\max\left\{ \sum_{\alpha \in \mathcal{A}} y_\alpha x_\alpha - h(x) : \sum_{\alpha \in \mathcal{A}} x_\alpha = 1 \text{ and } \forall \alpha \in \mathcal{A} : x_\alpha \geq 0 \right\}$$

By solving the Karush–Kuhn–Tucker (KKT) conditions to this optimization problem we readily get that $x$ lies in the interior of $\mathcal{X}$, since $\theta(0^+) = -\infty$, and thus we obtain that at the solution, it holds $y_\alpha = \theta'(x_\alpha) + \lambda$ for $\lambda \in \mathbb{R}$. Therefore, we have:

$$y_{\alpha,n} - y_{\alpha^*,n} = \theta'(x_{\alpha,n}) - \theta'(x_{\alpha^*,n}) \tag{A.2}$$

or equivalently:

$$\theta'(x_{\alpha,n}) = \theta'(x_{\alpha^*,n}) + y_{\alpha,n} - y_{\alpha^*,n} \leq \theta'(1) + y_{\alpha,n} - y_{\alpha^*,n} \tag{A.3}$$

Now, assume that there exists $\alpha \in \mathcal{A}$ such that $x_{\alpha,n}$ does not converge to 0, that is, $\limsup_n x_{\alpha,n} > \varepsilon$ for some $\varepsilon > 0$. Then, since $\theta$ is strongly convex, $\theta'$ is strictly increasing, and thus $\theta'(x_{\alpha,n}) \geq \theta'(\varepsilon)$ infinitely often. However, by taking $n \to \infty$ in [(A.3)](#), it implies that $\theta'(x_{\alpha,n}) \to -\infty$, which is a contradiction. Therefore, we conclude that for all $\alpha \neq \alpha^*$, it holds that $\lim_{n\to\infty} x_{\alpha,n} = 0$, and the convergence result follows.

Finally, note that since $\theta'$ is strictly increasing, it is invertible and its inverse is strictly increasing as well. Thus, for each $\alpha \neq \alpha^*$ we have:

$$x_{\alpha,n} \leq (\theta')^{-1}\big(\theta'(1) + y_{\alpha,n} - y_{\alpha^*,n}\big) \tag{A.4}$$

Therefore,

$$\|x_n - x^*\|_\infty = 1 - x_{\alpha^*,n} = \sum_{\alpha \neq \alpha^*} x_{\alpha,n} \leq \sum_{\alpha \neq \alpha^*} (\theta')^{-1}\big(\theta'(1) + y_{\alpha,n} - y_{\alpha^*,n}\big) \tag{A.5}$$

and our proof is complete. ∎

## B  Proofs for Continuous Time Algorithms

In this appendix, we provide the proof of [Theorem 1](#) and discuss the convergence of [(FTXL-D)](#) under a *non-vanishing* friction coefficient – that is, $r\dot{y}$ instead of $(r/t)\dot{y}$. First, we provide a lemma that is necessary for our analysis.

**Lemma B.1.** *Let $x^* = (\alpha_1^*, \ldots, \alpha_N^*) \in \mathcal{X}$ be a strict Nash equilibrium of $\Gamma$, and let $d$ denote the minimum payoff difference at equilibrium, i.e.,*

$$d := \min_{i \in \mathcal{N}} \min_{\beta_i \notin \mathrm{supp}(x_i^*)} [u_i(x_i^*; x_{-i}^*) - u_i(\beta_i; x_{-i}^*)] . \tag{B.1}$$

*Then, for any $c \in (0, d)$, there exists $M > 0$ such that if $y_{i\alpha_i^*} - y_{i\alpha_i} > M$ for all $\alpha_i \neq \alpha_i^* \in \mathcal{A}_i$ and $i \in \mathcal{N}$, then*

$$v_{i\alpha_i^*}(Q(y)) - v_{i\alpha_i}(Q(y)) > c \quad \text{for all } \alpha_i \neq \alpha_i^* \in \mathcal{A}_i, \text{ and } i \in \mathcal{N} . \tag{B.2}$$

*Proof.* Since $x^*$ is a strict Nash equilibrium, the minimum payoff difference $d$ at $x^*$ is bounded away from zero. Then, by continuity of the function $x \mapsto v(x)$, there exists a neighborhood $\mathcal{U}_*$ of $x^*$ such that for any $x \in \mathcal{U}_*$, it holds

$$v_{i\alpha_i^*}(x) - v_{i\alpha_i}(x) > c \quad \text{for all } \alpha_i \neq \alpha_i^* \in \mathcal{A}_i, \text{ and } i \in \mathcal{N} \tag{B.3}$$

Finally, by Giannou et al. [18, Lemma C.2.], there exists $M > 0$, such that $Q(y) \in \mathcal{U}_*$ for all $y \in \mathcal{V}^*$ with

$$y_{i\alpha_i^*} - y_{i\alpha_i} > M \quad \text{for all } \alpha_i \neq \alpha_i^* \in \mathcal{A}_i, \text{ and } i \in \mathcal{N} \tag{B.4}$$

Therefore, we readily get that if $y \in \mathcal{V}^*$ satisfies the above relation, then

$$v_{i\alpha_i^*}(Q(y)) - v_{i\alpha_i}(Q(y)) > c \quad \text{for all } \alpha_i \neq \alpha_i^* \in \mathcal{A}_i, \text{ and } i \in \mathcal{N} . \quad \blacksquare$$

We are now in a position to prove Theorem 1, which we restate below for convenience.

**Theorem 1.** *Let $x^*$ be a strict Nash equilibrium of $\Gamma$, and let $x(t) = Q(y(t))$ be a solution orbit of (FTXL-D). If $x(0)$ is sufficiently close to $x^*$, then $x(t)$ converges to $x^*$; in particular, if (FTXL-D) is run with logit best responses (that is, $Q \leftarrow \Lambda$), we have*

$$\|x(t) - x^*\|_\infty \leq \exp\left(C - \frac{ct^2}{2(r+1)}\right) \tag{6}$$

*where $C > 0$ is a constant that depends only on the initialization of (FTXL-D) and*

$$c = \frac{1}{2} \min_{i \in \mathcal{N}} \min_{\beta_i \notin \mathrm{supp}(x_i^*)} [u_i(x_i^*; x_{-i}^*) - u_i(\beta_i; x_{-i}^*)] > 0 \tag{7}$$

*is the minimum payoff difference at equilibrium.*

*Proof.* First of all, since $x^*$ is a strict Nash equilibrium, by Lemma B.1 for

$$c = \frac{1}{2} \min_{i \in \mathcal{N}} \min_{\beta_i \notin \mathrm{supp}(x_i^*)} [u_i(x_i^*; x_{-i}^*) - u_i(\beta_i; x_{-i}^*)]$$

there exists $M > 0$ such that if $y_{i\alpha_i^*} - y_{i\alpha_i} > M$ for all $\alpha_i \neq \alpha_i^* \in \mathcal{A}_i$ and $i \in \mathcal{N}$, then

$$v_{i\alpha_i^*}(Q(y)) - v_{i\alpha_i}(Q(y)) > c \quad \text{for all } \alpha_i \neq \alpha_i^* \in \mathcal{A}_i, \text{ and } i \in \mathcal{N} . \tag{B.5}$$

From now on, for notational convenience, we focus on player $i \in \mathcal{N}$ and drop the player-specific indices altogether. Then, for $\alpha \neq \alpha^* \in \mathcal{A}$, we let $z_\alpha(t) := y_\alpha(t) - y_\alpha^*(t)$, which evolves as:

$$\ddot{z}(t) = v_\alpha(x(t)) - v_{\alpha^*}(x(t)) - \frac{r}{t}\dot{z}_\alpha(t) \tag{B.6}$$

Let $y(0)$ such that $z_\alpha(0) = -M - \varepsilon$, for all $\alpha \neq \alpha^* \in \mathcal{A}$, where $\varepsilon > 0$ small. We will, first, show that $z(t) < -M$ for all $t \geq 0$. For the sake of contradiction, and denoting $T_0 := \inf\{t \geq 0 : z(t) \geq -M\}$, suppose that $T_0 < \infty$. Then, we readily get that for all $t < T_0$, it holds

$$v_\alpha(x(t)) - v_{\alpha^*}(x(t)) < -c \tag{B.7}$$

and therefore, for all $t \leq T_0$:

$$\ddot{z}_\alpha(t)t^r + rt^{r-1}\dot{z}_\alpha(t) = t^r [v_\alpha(x) - v_{\alpha^*}(x)] \leq -ct^r \tag{B.8}$$

which can be rewritten as:

$$\frac{d}{dt}(\dot{z}_\alpha(t)t^r) \le -ct^r \tag{B.9}$$

Integrating over $t < T_0$, we obtain $\dot{z}_\alpha(t)t^r \le -ct^{r+1}/(r+1)$, which readily implies:

$$z_\alpha(t) \le z_\alpha(0) - \frac{c}{2(r+1)}t^2$$

$$< -M - \frac{c}{2(r+1)}t^2 \tag{B.10}$$

By sending $t \to T_0$, we arrive at a contradiction. Therefore $z_\alpha(t) < -M$ for all $t \ge 0$, and the previous equation implies that for all $t \ge 0$ :

$$z_\alpha(t) \le z_\alpha(0) - \frac{c}{2(r+1)}t^2 \tag{B.11}$$

and invoking Lemma A.1, we get the convergence result. Finally, translating the score-differences to the primal space $\mathcal{X}$, we get:

$$\|x(t) - x^*\|_\infty = \max_{i \in \mathcal{N}}\{1 - x_{i\alpha_i^*}(t)\} \tag{B.12}$$

For the case of logit best responses, i.e., when $Q \leftarrow \Lambda$, and assuming that the maximum above is attained for player $i \in \mathcal{N}$, we obtain

$$\|x(t) - x^*\|_\infty = \frac{\sum_{\alpha_i \ne \alpha_i^*} \exp(z_{\alpha_i}(t))}{1 + \sum_{\alpha_i \ne \alpha_i^*} \exp(z_{\alpha_i}(t))}$$

$$\le \sum_{\alpha_i \ne \alpha_i^*} \exp(z_{\alpha_i}(t))$$

$$\le |\mathcal{A}_i| \exp\left(z_{\alpha_i}(0) - \frac{c}{2(r+1)}t^2\right)$$

$$\le \exp\left(C - \frac{c}{2(r+1)}t^2\right) \tag{B.13}$$

for $C = \log|\mathcal{A}_i| + z_{\alpha_i}(0)$. ∎

Now, moving to the case where we use a constant friction coefficient $-r\dot{y}$ instead of $(r/t)\dot{y}$, (FTXL-D) becomes:

$$\frac{d^2y}{dt^2} = v(Q(y)) - r\frac{dy}{dt} \tag{B.14}$$

Under, (B.14), we obtain the following convergence result.

**Theorem B.1.** *Let $x^*$ be a strict Nash equilibrium of $\Gamma$, and let $x(t) = Q(y(t))$ be a solution orbit of (B.14). If $x(0)$ is sufficiently close to $x^*$, then $x(t)$ converges to $x^*$; in particular, if (B.14) is run with logit best responses (that is, $Q \leftarrow \Lambda$), we have*

$$\|x(t) - x^*\|_\infty \le \exp\left(C - \frac{c}{r}t - \frac{c}{r^2}e^{-rt} + \frac{c}{r^2}\right) \tag{B.15}$$

*where $C > 0$ is a constant that depends on the initialization of (B.14) and*

$$c = \frac{1}{2}\min_{i \in \mathcal{N}}\min_{\beta_i \notin \text{supp}(x_i^*)}[u_i(x_i^*; x_{-i}^*) - u_i(\beta_i; x_{-i}^*)] > 0 \tag{B.16}$$

*is the minimum payoff difference at equilibrium.*

*Proof.* The initial steps of proof of Theorem B.1 are similar to the proof of Theorem 1, which we include for the sake of completeness.

Specifically, by Lemma B.1 there exists $M > 0$ such that if $y_{i\alpha_i^*} - y_{i\alpha_i} > M$ for all $\alpha_i \ne \alpha_i^* \in \mathcal{A}_i$ and $i \in \mathcal{N}$, then

$$v_{i\alpha_i^*}(Q(y)) - v_{i\alpha_i}(Q(y)) > c \quad \text{for all } \alpha_i \ne \alpha_i^* \in \mathcal{A}_i, \text{ and } i \in \mathcal{N}. \tag{B.17}$$

Now, for notational convenience, we focus on player $i \in \mathcal{N}$ and drop the player-specific indices altogether. Then, for $\alpha \neq \alpha^* \in \mathcal{A}$, we let $z_\alpha(t) := y_\alpha(t) - y_\alpha^*(t)$, which evolves as:

$$\ddot{z}(t) = v_\alpha(x(t)) - v_{\alpha^*}(x(t)) - r\dot{z}_\alpha(t) \tag{B.18}$$

Let $y(0)$ such that $z_\alpha(0) = -M - \varepsilon$, for all $\alpha \neq \alpha^* \in \mathcal{A}$, where $\varepsilon > 0$ small. As in the proof of Theorem 1, we will, first, show that $z(t) < -M$ for all $t \geq 0$. For the sake of contradiction, and denoting $T_0 := \inf\{t \geq 0 : z(t) \geq -M\}$, suppose that $T_0 < \infty$. Then, we readily get that for all $t < T_0$, it holds

$$v_\alpha(x(t)) - v_{\alpha^*}(x(t)) < -c \tag{B.19}$$

and therefore, for all $t \leq T_0$:

$$\ddot{z}_\alpha(t)e^{rt} + re^{rt}\dot{z}_\alpha(t) = e^{rt}\left[v_\alpha(x) - v_{\alpha^*}(x)\right] \leq -ce^{rt} \tag{B.20}$$

which can be rewritten as:

$$\frac{d}{dt}\left(\dot{z}_\alpha(t)e^{rt}\right) \leq -ce^{rt} \tag{B.21}$$

Integrating over $t < T_0$, and using that $\dot{z}_\alpha(0) = 0$, we obtain $\dot{z}_\alpha(t) \leq -c/r + ce^{-rt}/r$, which implies:

$$
\begin{aligned}
z_\alpha(t) &\leq z_\alpha(0) - \frac{c}{r}t - \frac{c}{r^2}e^{-rt} + \frac{c}{r^2} \\
&= z_\alpha(0) - \frac{c}{r^2}\left(rt + e^{-rt} - 1\right) \\
&< z_\alpha(0) \\
&< -M
\end{aligned}
\tag{B.22}
$$

where we used the fact that $x + e^{-x} - 1 \geq 0$ for all $x \in \mathbb{R}$ with equality if and only if $x = 0$. By sending $t \to T_0$, we arrive at a contradiction. Therefore $z_\alpha(t) < -M$ for all $t \geq 0$, and the previous equation implies that for all $t \geq 0$ :

$$z_\alpha(t) \leq z_\alpha(0) - \frac{c}{r}t - \frac{c}{r^2}e^{-rt} + \frac{c}{r^2} \tag{B.23}$$

and invoking Lemma A.1 for $\theta(x) = x\log x$, we get the convergence result. ∎

## C    Proofs for discrete-time algorithms with full information

In this section, we provide the results for the (FTXL) algorithm with full-information feedback. First, we discuss the rates obtained by the direct discretization of (FTXL-D) with both vanishing and non-vanishing friction, and then provide the proof of Theorem 2, our main result, for the full-information case.

**C.1. FTXL with vanishing friction.** First, we provide the rate of convergence for the discrete version of (FTXL-D) with vanishing friction:

$$
\begin{aligned}
p_{i,n+1} &= p_{i,n}\left(1 - \frac{\gamma r}{n}\right) + \gamma\hat{v}_{i,n} \\
y_{i,n+1} &= y_{i,n} + \gamma p_{i,n+1}
\end{aligned}
\tag{C.1}
$$

To streamline our presentation, we consider the setup of Example 3.1 that provides a lower bound for the algorithm.

**Proposition C.1.** *Consider the single-player game $\Gamma$ with actions* A *and* B *such that $u(\mathtt{A}) - u(\mathtt{B}) = 1$ of Example 3.1, and let $x_n = \Lambda(y_n)$ be the sequence of play generated by (C.1). Then, denoting by $x^* = (1,0)$ the strict Nash equilibrium, we have:*

$$\|x_T - x^*\|_\infty \sim \exp\left(C - \frac{\gamma^2 T^2}{2(\gamma r + 1)}\right). \tag{C.2}$$

*where $C > 0$ is a constant that depends only on the initialization of the algorithm.*

*Proof.* We first define the score-difference

$$w_n := p_{\mathrm{B},n} - p_{\mathrm{A},n} \tag{C.3}$$

with initial condition $w_1 = 0$. Then, unfolding according to the sequence of play, we obtain:

$$\begin{aligned}
w_{n+1} &= w_n\left(1 - \frac{\gamma r}{n}\right) + \gamma(u(\mathrm{B}) - u(\mathrm{A})) \\
&= w_n\left(1 - \frac{\gamma r}{n}\right) - \gamma \\
&= -\gamma \sum_{k=1}^{n-1} \prod_{\ell=0}^{k-1}\left(1 - \frac{\gamma r}{n - \ell}\right) - \gamma
\end{aligned} \tag{C.4}$$

We next define for $n \in \mathbb{N}$ the difference $z_n := y_{\mathrm{B},n} - y_{\mathrm{A},n}$. Thus, unfolding it, we obtain:

$$\begin{aligned}
z_{n+1} &= z_n + \gamma w_{n+1} \\
&= z_n - \gamma^2\left(1 + \sum_{k=1}^{n-1} \prod_{\ell=0}^{k-1}\left(1 - \frac{\gamma r}{n - \ell}\right)\right) \\
&= z_1 - \gamma^2 \sum_{m=1}^{n}\left(1 + \sum_{k=1}^{m-1} \prod_{\ell=0}^{k-1}\left(1 - \frac{\gamma r}{m - \ell}\right)\right)
\end{aligned} \tag{C.5}$$

Now, using Lemma C.1, which we provide after this proof, we obtain that

$$\begin{aligned}
z_{n+1} &= z_1 - \gamma^2 \sum_{m=1}^{n}\left(1 + \frac{m - \gamma r}{1 + \gamma r} - \frac{1}{1 + \gamma r}\prod_{\ell=1}^{m}\left(1 - \frac{\gamma r}{\ell}\right)\right) \\
&= z_1 - \gamma^2 \frac{n(n+1)}{2(1 + \gamma r)} - \gamma^2 n\left(1 - \frac{\gamma r}{1 + \gamma r}\right) + \frac{\gamma^2}{1 + \gamma r}\sum_{m=1}^{n}\prod_{\ell=1}^{m}\left(1 - \frac{\gamma r}{\ell}\right) \\
&= z_1 - \frac{\gamma^2 n^2}{2(1 + \gamma r)} + \Theta(n)
\end{aligned} \tag{C.6}$$

and invoking Lemma A.1 for $\theta(x) = x \log x$, we get the result. ∎

The following lemma is a necessary tool for obtaining the exact convergence rate in Proposition C.2.

**Lemma C.1.** *For any $m \in \mathbb{N}$ and $a > 0$, we have that*

$$\sum_{k=1}^{m-1} \prod_{\ell=0}^{k-1}\left(1 - \frac{a}{m - \ell}\right) = \frac{m - a}{1 + a} - \frac{1}{1 + a}\prod_{\ell=1}^{m}\left(1 - \frac{a}{\ell}\right) \tag{C.7}$$

*Proof.* First, by expanding the inner product, we can rewrite the expression as

$$\begin{aligned}
\sum_{k=1}^{m-1} \prod_{\ell=0}^{k-1}\left(1 - \frac{a}{m - \ell}\right) &= \sum_{k=1}^{m-1} \prod_{\ell=0}^{k-1}\left(\frac{m - \ell - a}{m - \ell}\right) \\
&\qquad \sum_{k=1}^{m-1} \frac{(m - a)\dots(m - k + 1 - a)}{m \dots (m - k + 1)} \\
&= \sum_{k=1}^{m-1} \frac{(m - a)!(m - k)!}{(m - k - a)!m!} \\
&= \frac{(m - a)!}{m!} \sum_{k=1}^{m-1} \frac{(m - k)!}{(m - k - a)!}
\end{aligned} \tag{C.8}$$

where with a slight abuse of notation we use the factorial notation $(m - a)!$ to denote the Gamma function evaluated at $m - a + 1$, i.e., $\Gamma(m - a + 1)$.

Now, defining the quantity

$$F_m := \frac{(m - a)!}{m!} \sum_{k=1}^{m} \frac{(m - k)!}{(m - k - a)!}$$

the difference of two consecutive terms evolves as:

$$F_{m+1} - F_m = \frac{(m+1-a)!}{(m+1)!} \sum_{k=1}^{m+1} \frac{(m+1-k)!}{(m+1-k-a)!} - \frac{(m-a)!}{m!} \sum_{k=1}^{m} \frac{(m-k)!}{(m-k-a)!}$$

$$= \frac{m+1-a}{m+1} + \frac{(m+1-a)!}{(m+1)!} \sum_{k=2}^{m+1} \frac{(m+1-k)!}{(m+1-k-a)!} - \frac{(m-a)!}{m!} \sum_{k=1}^{m} \frac{(m-k)!}{(m-k-a)!}$$

$$= \frac{m+1-a}{m+1} + \frac{(m+1-a)!}{(m+1)!} \sum_{k=2}^{m+1} \frac{(m+1-k)!}{(m+1-k-a)!} - \frac{(m-a)!}{m!} \sum_{k=2}^{m+1} \frac{(m-k+1)!}{(m-k+1-a)!}$$

$$= \frac{m+1-a}{m+1} + \sum_{k=2}^{m+1} \frac{(m+1-a)!(m+1-k)! - (m+1)(m-a)!(m-k+1)!}{(m+1)!(m+1-a-k)!}$$

$$= \frac{m+1-a}{m+1} + \sum_{k=2}^{m+1} \frac{(m-a)!(m+1-k)!(m+1-a-m-1)}{(m+1)!(m+1-a-k)!}$$

$$= \frac{m+1-a}{m+1} - a \sum_{k=2}^{m+1} \frac{(m-a)!(m+1-k)!}{(m+1)!(m+1-a-k)!}$$

$$= \frac{m+1-a}{m+1} - \frac{a}{m+1-a} \left[ \sum_{k=1}^{m+1} \frac{(m+1-k)!(m+1-a)!}{(m+1)!(m+1-a-k)!} - \frac{m+1-a}{m+1} \right]$$

$$= 1 - \frac{a}{m+1-a} F_{m+1} \tag{C.9}$$

Thus, we readily obtain the recurrence relation

$$\frac{m+1}{m+1-a} F_{m+1} = F_m + 1 . \tag{C.10}$$

We continue the proof by induction. To this end, we will show that

$$F_m = \frac{m-a}{1+a} + \frac{a}{1+a} \prod_{\ell=1}^{m} \frac{\ell-a}{\ell} . \tag{C.11}$$

For the base case, note that

$$F_1 = (1-a) = \frac{1-a}{1+a} + \frac{a}{1+a}(1-a) \tag{C.12}$$

For the inductive step, suppose that (C.11) holds for $m \in \mathbb{N}$. Then, we have:

$$\frac{m+1}{m+1-a} F_{m+1} = \frac{m-a}{1+a} + \frac{a}{1+a} \prod_{\ell=1}^{m} \left( \frac{\ell-a}{\ell} \right) + 1$$

$$= \frac{m+1}{1+a} + \frac{a}{1+a} \prod_{\ell=1}^{m} \frac{\ell-a}{\ell} \tag{C.13}$$

which implies the inductive step

$$F_{m+1} = \frac{m+1-a}{1+a} + \frac{a}{1+a} \prod_{\ell=1}^{m+1} \frac{\ell-a}{\ell} \tag{C.14}$$

and thus (C.11) holds for all $m \in \mathbb{N}$. Finally, to complete the proof notice that

$$\sum_{k=1}^{m-1} \prod_{\ell=0}^{k-1} (1 - \frac{a}{m-\ell}) = \frac{(m-a)!}{m!} \sum_{k=1}^{m-1} \frac{(m-k)!}{(m-k-a)!}$$

$$= F_m - \prod_{\ell=0}^{m-1} \left( 1 - \frac{a}{m-\ell} \right)$$

$$= \frac{m-a}{1+a} + \frac{a}{1+a} \prod_{\ell=1}^{m} \frac{\ell-a}{\ell} - \prod_{\ell=1}^{m} \left( 1 - \frac{a}{\ell} \right)$$

$$= \frac{m-a}{1+a} - \frac{1}{1+a} \prod_{\ell=1}^{m} \left( 1 - \frac{a}{\ell} \right) \tag{C.15}$$

as was to be shown. ∎

Next, we discuss the cases of non-vanishing and zero friction.

**C.2. FTXL with non-vanishing friction.** We continue this section by considering the case of non-vanishing friction in analogy to the continuous-time case, as per Appendix B. Specifically, we consider the discrete version of (FTXL-D) with non-vanishing friction, as follows:

$$p_{i,n+1} = p_{i,n}(1 - \gamma r) + \gamma \hat{v}_{i,n}$$
$$y_{i,n+1} = y_{i,n} + \gamma p_{i,n+1}$$

(C.16)

with $\gamma r < 1$. Below, we provide the rate of convergence for the setup of *Example* 3.1, as we did before. Namely, we obtain a linear convergence rate, as the following proposition suggests.

**Proposition C.2.** *Consider the single-player game $\Gamma$ with actions $\mathtt{A}$ and $\mathtt{B}$ such that $u(\mathtt{A}) - u(\mathtt{B}) = 1$ of Example 3.1, and let $x_n = \Lambda(y_n)$ be the sequence of play generated by (C.16). Then, denoting by $x^* = (1,0)$ the strict Nash equilibrium, we have:*

$$\|x_n - x^*\|_\infty \sim \exp\left(C - \frac{\gamma}{r}n\right).$$

(C.17)

*where $C > 0$ is a constant that depends on the initialization of the algorithm.*

*Proof.* We first define the score-difference

$$w_n := p_{\mathtt{B},n} - p_{\mathtt{A},n}$$

(C.18)

with initial condition $w_1 = 0$. Then, unfolding according to the sequence of play, we obtain:

$$
\begin{aligned}
w_{n+1} &= w_n(1 - \gamma r) + \gamma(u(\mathtt{B}) - u(\mathtt{A})) \\
&= w_n(1 - \gamma r) - \gamma \\
&= \ldots \\
&= -\gamma \sum_{k=0}^{n-1}(1 - \gamma r)^k \\
&= -\frac{1 - (1 - \gamma r)^n}{r}
\end{aligned}
$$

(C.19)

We next define for $n \in \mathbb{N}$ the difference $z_n := y_{\mathtt{B},n} - y_{\mathtt{A},n}$. Thus, unfolding it, we obtain:

$$
\begin{aligned}
z_{n+1} &= z_n + \gamma w_{n+1} \\
&= z_n - \gamma \frac{1 - (1 - \gamma r)^n}{r} \\
&= z_1 - \gamma \sum_{m=1}^{n} \frac{1 - (1 - \gamma r)^m}{r} \\
&= z_1 - \frac{\gamma}{r}\left(n - (1 - \gamma r)\frac{1 - (1 - \gamma r)^n}{\gamma r}\right) \\
&= z_1 - \frac{\gamma}{r}n + \mathcal{O}(1)
\end{aligned}
$$

(C.20)

and invoking Lemma A.1 for $\theta(x) = x \log x$, we get the result. ∎

**C.3. FTXL with zero friction.** Moving forward to the case of $r = 0$ as presented in Section 4, we provide the proof of Theorem 2, which we restate below for convenience.

**Theorem 2.** *Let $x^*$ be a strict Nash equilibrium of $\Gamma$, and let $x_n = Q(y_n)$ be the sequence of play generated by (FTXL) with full information feedback of the form (11a). If $x_1$ is initialized sufficiently close to $x^*$, then $x_n$ converges to $x^*$; in particular, if (FTXL) is run with logit best responses (that is, $Q \leftarrow \Lambda$), we have*

$$\|x_T - x^*\|_\infty \leq \exp\left(C - c\gamma^2\frac{T(T-1)}{2}\right) = \exp\left(-\Theta(T^2)\right)$$

(12)

*where $C > 0$ is a constant that depends only on the initialization of (FTXL) and*

$$c = \frac{1}{2} \min_{i \in \mathcal{N}} \min_{\beta_i \notin \text{supp}(x_i^*)} [u_i(x_i^*; x_{-i}^*) - u_i(\beta_i; x_{-i}^*)] > 0$$

(13)

*is the minimum payoff difference at equilibrium.*

*Proof.* First of all, since $x^*$ is a strict Nash equilibrium, by Lemma B.1 for

$$c = \frac{1}{2} \min_{i \in \mathcal{N}} \min_{\beta_i \notin \text{supp}(x_i^*)} [u_i(x_i^*; x_{-i}^*) - u_i(\beta_i; x_{-i}^*)]$$

there exists $M > 0$ such that if $y_{i\alpha_i^*} - y_{i\alpha_i} > M$ for all $\alpha_i \neq \alpha_i^* \in \mathcal{A}_i$ and $i \in \mathcal{N}$, then

$$v_{i\alpha_i^*}(Q(y)) - v_{i\alpha_i}(Q(y)) > c \quad \text{for all } \alpha_i \neq \alpha_i^* \in \mathcal{A}_i, \text{ and } i \in \mathcal{N} . \tag{C.21}$$

For notational convenience, we focus on player $i$ and drop the player-specific indices altogether. Let $\alpha \neq \alpha^* \in \mathcal{A}$, and define for $n \in \mathbb{N}$ the quantities $w_{\alpha,n}$ and $z_{\alpha,n}$ as

$$w_{\alpha,n} := \langle p_n, e_\alpha - e_\alpha^* \rangle, \qquad z_{\alpha,n} := \langle y_n, e_\alpha - e_\alpha^* \rangle \tag{C.22}$$

where $e_\alpha, e_\alpha^*$ are the standard basis vectors corresponding to $\alpha, \alpha^* \in \mathcal{A}$.

Let initial conditions $y_1$ such that $y_{\alpha,1} - y_{\alpha^*,1} = -M - \varepsilon$, for all $\alpha \neq \alpha^* \in \mathcal{A}$, where $\varepsilon > 0$ small, and $p_1 = 0$. We will first show by induction that $z_{\alpha,n} < -M$ for all $n \in \mathbb{N}$. To this end, unfolding the recursion, we obtain:

$$\begin{aligned}
w_{\alpha,n+1} &= w_{\alpha,n} + \gamma \langle \hat{v}_n, e_\alpha - e_\alpha^* \rangle \\
&= w_{\alpha,n} + \gamma \langle v(x_n), e_\alpha - e_\alpha^* \rangle \\
&= \gamma \sum_{k=1}^{n} \langle v(x_k), e_\alpha - e_\alpha^* \rangle
\end{aligned} \tag{C.23}$$

where we used that $w_1 = 0$. Now, for the sake of induction, suppose that

$$z_{\alpha,k} < -M \quad \text{for all } k = 1, \dots, n \tag{C.24}$$

which implies that $\langle v(x_k), e_\alpha - e_\alpha^* \rangle < -c$. With this in hand, we will prove that $z_{\alpha,n+1} < -M$, as well. Specifically, we have:

$$\begin{aligned}
z_{\alpha,n+1} = z_{\alpha,n} + \gamma w_{\alpha,n+1} &= z_{\alpha,n} + \gamma^2 \sum_{k=1}^{n} \langle v(x_k), e_\alpha - e_\alpha^* \rangle \\
&\leq z_{\alpha,n} - c\gamma^2 n \\
&\leq z_{\alpha,1} - c\gamma^2 \sum_{\ell=1}^{n} \ell \\
&< -M
\end{aligned} \tag{C.25}$$

where we used the inductive hypothesis and the initial condition. Therefore, we conclude by induction that $z_{\alpha,n} < -M$ for all $n \in \mathbb{N}$. Thus, we readily obtain that after $T$ time-step:

$$z_T \leq z_{\alpha,1} - c\gamma^2 \sum_{\ell=1}^{T-1} \ell \leq z_{\alpha,1} - c\gamma^2 \frac{T(T-1)}{2} \tag{C.26}$$

and invoking Lemma A.1 for $\theta(x) = x \log x$, we get the result. $\blacksquare$

## D   Proofs for discrete-time algorithms with partial information

In this appendix, we provide the proofs of Theorem 3 and Theorem 4 that correspond to the convergence of (FTXL) with realization-based and bandit feedback, respectively. For this, we need the following lemma, which provides a maximal bound on a martingale process. Namely, we have:

**Lemma D.1.** *Let $M_n := \sum_{k=1}^{n} \gamma_k \xi_k$ be a martingale with respect to $(\mathcal{F}_n)_{n \in \mathbb{N}}$ with $\mathbb{E}[\|\xi_n\|_*^q] \leq \sigma_n^q$ for some $q > 2$. Then, for $\mu \in (0, 1)$ and $n \in \mathbb{N}$:*

$$\mathbb{P}\left( \sup_{k \leq n} |M_k| > c \left( \sum_{k=1}^{n} \gamma_k \right)^\mu \right) \leq A_q \frac{\sum_{k=1}^{n} \gamma_k^{q/2+1} \sigma_k^q}{\left( \sum_{k=1}^{n} \gamma_k \right)^{1+q(\mu-1/2)}} \tag{D.1}$$

*where $A_q$ is a constant depending only on $c$ and $q$.*

*Proof.* Fix some $\mu \in (0, 1)$. By Doob's maximal inequality [20, Corollary 2.1], we have:

$$\mathbb{P}\left(\sup_{k \leq n} |M_k| > c\left(\sum_{k=1}^n \gamma_k\right)^{\mu}\right) \leq \frac{\mathbb{E}[|M_n|^q]}{c^q \left(\sum_{k=1}^n \gamma_k\right)^{q\mu}} \tag{D.2}$$

Now, applying the Burkholder–Davis–Gundy inequality [20, Theorem 2.10], we get that

$$\mathbb{E}[|M_n|^q] \leq A_q \mathbb{E}\left[\left(\sum_{k=1}^n \gamma_k^2 \|\xi_k\|_*^2\right)^{q/2}\right] \tag{D.3}$$

where $A_q$ is a constant depending only on $c$ and $q$. Now, we will invoke the generalized Hölder's inequality [4], we have:

$$\left(\sum_{k=1}^n a_k b_k\right)^{\rho} \leq \left(\sum_{k=1}^n a_k^{\frac{\lambda\rho}{\rho-1}}\right)^{\rho-1} \sum_{k=1}^n a_k^{(1-\lambda)\rho} b_k^{\rho} \tag{D.4}$$

for $a_k, b_k \geq 0$, $\rho > 1$ and $\lambda \in [0, 1)$. Thus, setting $a_k = \gamma_k^2$, $b_k = \|\xi_k\|_*^2$, $\rho = q/2$ and $\lambda = 1/2 - 1/q$, (D.2), combined with (D.3), becomes:

$$\mathbb{P}\left(\sup_{k \leq n} |M_k| > c\left(\sum_{k=1}^n \gamma_k\right)^{\mu}\right) \leq A_q \frac{\left(\sum_{k=1}^n \gamma_k\right)^{q/2-1} \sum_{k=1}^n \gamma_k^{q/2+1} \mathbb{E}[\|\xi_k\|_*^q]}{\left(\sum_{k=1}^n \gamma_k\right)^{q\mu}}$$

$$\leq A_q \frac{\sum_{k=1}^n \gamma_k^{q/2+1} \sigma_k^q}{\left(\sum_{k=1}^n \gamma_k\right)^{1+q(\mu-1/2)}} \tag{D.5}$$

and our proof is complete. ∎

With this tool in hand, we proceed to prove the convergence of (FTXL) under realization-based feedback. For convenience, we restate the relevant result below.

**Theorem 3.** *Let $x^*$ be a strict Nash equilibrium of $\Gamma$, fix some confidence level $\delta > 0$, and let $x_n = Q(y_n)$ be the sequence of play generated by (FTXL) with realization-based feedback as per (11b) and a sufficiently small step-size $\gamma > 0$. Then there exists a neighborhood $\mathcal{U}$ of $x^*$ such that*

$$\mathbb{P}(x_n \to x^* \text{ as } n \to \infty) \geq 1 - \delta \qquad \text{if } x_1 \in \mathcal{U}. \tag{14}$$

*In particular, if* (FTXL) *is run with logit best responses (that is, $Q \leftarrow \Lambda$), there exist positive constants $C, c > 0$ as in [Theorem 2](#) such that on the event $\{x_n \to x^* \text{ as } n \to \infty\}$:*

$$\|x_T - x^*\|_{\infty} \leq \exp\left(C - c\gamma^2 \frac{T(T-1)}{2} + \frac{3}{5} c\gamma^{5/3} T^{5/3}\right) = \exp\left(-\Theta(T^2)\right). \tag{15}$$

*Proof.* First of all, since $x^*$ is a strict Nash equilibrium, by [Lemma B.1](#) for

$$c = \frac{1}{2} \min_{i \in \mathcal{N}} \min_{\beta_i \notin \text{supp}(x_i^*)} [u_i(x_i^*; x_{-i}^*) - u_i(\beta_i; x_{-i}^*)]$$

there exists $M > 0$ such that if $y_{i\alpha_i^*} - y_{i\alpha_i} > M$ for all $\alpha_i \neq \alpha_i^* \in \mathcal{A}_i$ and $i \in \mathcal{N}$, then

$$v_{i\alpha_i^*}(Q(y)) - v_{i\alpha_i}(Q(y)) > c \quad \text{for all } \alpha_i \neq \alpha_i^* \in \mathcal{A}_i, \text{ and } i \in \mathcal{N}. \tag{D.6}$$

For notational convenience, we focus on player $i$ and drop the player-specific indices altogether. Let $\alpha \neq \alpha^* \in \mathcal{A}$, and define for $n \in \mathbb{N}$ the quantities $w_{\alpha,n}$ and $z_{\alpha,n}$ as

$$w_{\alpha,n} := \langle p_n, e_{\alpha} - e_{\alpha}^* \rangle, \qquad z_{\alpha,n} := \langle y_n, e_{\alpha} - e_{\alpha}^* \rangle \tag{D.7}$$

where $e_{\alpha}, e_{\alpha}^*$ are the standard basis vectors corresponding to $\alpha, \alpha^* \in \mathcal{A}$.

Then, unfolding the recursion, we obtain:

$$w_{\alpha,n+1} = w_{\alpha,n} + \gamma\langle \hat{v}_n, e_{\alpha} - e_{\alpha}^* \rangle = w_{\alpha,n} + \gamma\langle v(x_n), e_{\alpha} - e_{\alpha}^* \rangle + \gamma\langle U_n, e_{\alpha} - e_{\alpha}^* \rangle$$

$$= \gamma \sum_{k=1}^n \langle v(x_k), e_{\alpha} - e_{\alpha}^* \rangle + \gamma \sum_{k=1}^n \langle U_k, e_{\alpha} - e_{\alpha}^* \rangle \tag{D.8}$$

where we used that $w_1 = 0$. Now, define the stochastic process $\{M_n\}_{n\in\mathbb{N}}$ as

$$M_n := \gamma \sum_{k=1}^{n} \langle U_k, e_\alpha - e_\alpha^* \rangle \tag{D.9}$$

which is a martingale, since $\mathbb{E}[U_n \mid \mathcal{F}_n] = 0$. Moreover, note that

$$\|U_n\|_* = \|v(\alpha_n) - v(x_n)\|_* \leq 2 \max_{\alpha\in\mathcal{A}} \|v(\alpha)\|_* \tag{D.10}$$

and, thus, we readily obtain that $\mathbb{E}[\|U_n\|_*^q \mid \mathcal{F}_n] \leq \sigma^q$ for $\sigma = 2\max_{\alpha\in\mathcal{A}}\|v(\alpha)\|_*$ and all $q \in [1,\infty]$.

By Lemma D.1 for $\gamma_n = \gamma$, $\sigma_n = \sigma$, $\xi_n = \langle U_n, e_\alpha - e_\alpha^* \rangle$, $c$ as in Theorem 2, and $\mu \in (0,1)$, $q > 2$ whose values will be determined next, there exists $A_q > 0$ such that:

$$\delta_n := \mathbb{P}\left(\sup_{k\leq n}|M_k| > c(\gamma n)^\mu\right) \leq A_q \sigma^q \frac{n\gamma^{q/2+1}}{(\gamma n)^{1+q(\mu-1/2)}}$$

$$\leq A_q \sigma^q \frac{\gamma^{q(1-\mu)}}{n^{q(\mu-1/2)}} \tag{D.11}$$

Now, we need to guarantee that there exist $\mu \in (0,1)$, $q > 2$, such that

$$\sum_{n=1}^{\infty} \delta_n < \infty \tag{D.12}$$

For this, we simply need $q(\mu - 1/2) > 1$, or equivalently, $\mu > 1/2 + 1/q$, which implies that $\mu \in (1/2, 1)$.

Therefore, for $\gamma$ small enough, we get $\sum_{n=1}^{\infty}\delta_n < \delta$, and therefore:

$$\mathbb{P}\left(\bigcap_{n=1}^{\infty}\left\{\sup_{k\leq n}|M_k| \leq c(\gamma n)^\mu\right\}\right) = 1 - \mathbb{P}\left(\bigcup_{n=1}^{\infty}\left\{\sup_{k\leq n}|M_k| > c(\gamma n)^\mu\right\}\right)$$

$$\geq 1 - \sum_{n=1}^{\infty}\delta_n$$

$$\geq 1 - \delta \tag{D.13}$$

From now on, we denote the good event $\bigcap_{n=1}^{\infty}\{\sup_{k\leq n}|M_k| \leq c(\gamma n)^\mu\}$ by $E$. Then, with probability at least $1 - \delta$:

$$w_{\alpha,n+1} \leq \gamma \sum_{k=1}^{n} \langle v(x_k), e_\alpha - e_\alpha^* \rangle + c(\gamma n)^\mu \quad \text{for all } n \in \mathbb{N}. \tag{D.14}$$

Furthermore, we have that for $n > N_0 := \lceil 1/\gamma \rceil$, we readily get that $\gamma n > (\gamma n)^\mu$. Therefore, setting

$$R := c\gamma \sum_{k=1}^{N_0-1} ((\gamma k)^\mu - \gamma k) \tag{D.15}$$

we obtain:

$$-c\gamma \sum_{k=1}^{n} (\gamma k - (\gamma k)^\mu) \leq R \tag{D.16}$$

for all $n \in \mathbb{N}$. Then, initializing $y_1$ such that $z_{\alpha,1} < -M - R$, we will show that $z_{\alpha,n} < -M$ for all $n \in \mathbb{N}$ with probability at least $1 - \delta$. For this, suppose that $E$ is realized, and assume that

$$z_{\alpha,k} < -M \quad \text{for all } k = 1, \ldots, n \tag{D.17}$$

We will show that $z_{\alpha,n+1} < -M$, as well. For this, we have:

$$z_{\alpha,n+1} = z_{\alpha,n} + \gamma w_{\alpha,n+1}$$

$$\leq z_{\alpha,n} + \gamma\left(\gamma \sum_{k=1}^{n}\langle v(x_k), e_\alpha - e_\alpha^* \rangle + c(\gamma n)^\mu\right)$$

$$\leq z_{\alpha,n} - c\gamma(\gamma n - (\gamma n)^\mu)$$

$$\leq z_{\alpha,1} - c\gamma \sum_{k=1}^{n}(\gamma k - (\gamma k)^\mu)$$

$$\leq -M - R - c\gamma \sum_{k=1}^{n}(\gamma k - (\gamma k)^\mu)$$

$$< -M \tag{D.18}$$

Therefore, we conclude by induction that $z_{\alpha,n} < -M$ for all $n \in \mathbb{N}$. Thus, we readily obtain that with probability at least $1 - \delta$ it holds:

$$z_{\alpha,T} \leq z_{\alpha,1} - c\gamma \sum_{k=1}^{T-1}(\gamma k - (\gamma k)^\mu)$$

$$\leq z_{\alpha,1} - c\gamma^2 \frac{T(T-1)}{2} + c\gamma^{1+\mu} \int_0^T t^\mu dt$$

$$\leq z_{\alpha,1} - c\gamma^2 \frac{T(T-1)}{2} + c\gamma^{1+\mu} \frac{T^{\mu+1}}{\mu+1} \tag{D.19}$$

for all $T \in \mathbb{N}$. Setting $\mu = 2/3$ and invoking Lemma A.1 for $\theta(x) = x \log x$, we get the result. ∎

Finally, we prove the convergence of (FTXL) with bandit feedback. Again, for convenience, we restate the relevant result below.

**Theorem 4.** *Let $x^*$ be a strict Nash equilibrium of $\Gamma$, fix some confidence level $\delta > 0$, and let $x_n = Q(y_n)$ be the sequence of play generated by (FTXL) with bandit feedback of the form (11c), an IWE exploration parameter $\varepsilon_n \propto 1/n^{\ell_\varepsilon}$ for some $\ell_\varepsilon \in (0, 1/2)$, and a sufficiently small step-size $\gamma > 0$. Then there exists a neighborhood $\mathcal{U}$ of $x^*$ in $\mathcal{X}$ such that*

$$\mathbb{P}(x_n \to x^* \text{ as } n \to \infty) \geq 1 - \delta \qquad \text{if } x_1 \in \mathcal{U}. \tag{17}$$

*In particular, if (FTXL) is run with logit best responses (that is, $Q \leftarrow \Lambda$), there exist positive constants $C, c > 0$ as in Theorem 2 such that on the event $\{x_n \to x^* \text{ as } n \to \infty\}$*

$$\|x_T - x^*\|_\infty \leq \exp\left(C - c\gamma^2 \frac{T(T-1)}{2} + \frac{5}{9}c\gamma^{9/5}T^{9/5}\right) = \exp\left(-\Theta(T^2)\right). \tag{18}$$

*Proof.* First of all, since $x^*$ is a strict Nash equilibrium, by Lemma B.1 for

$$c = \frac{1}{2} \min_{i \in \mathcal{N}} \min_{\beta_i \notin \mathrm{supp}(x_i^*)} [u_i(x_i^*; x_{-i}^*) - u_i(\beta_i; x_{-i}^*)]$$

there exists $M > 0$ such that if $y_{i\alpha_i^*} - y_{i\alpha_i} > M$ for all $\alpha_i \neq \alpha_i^* \in \mathcal{A}_i$ and $i \in \mathcal{N}$, then

$$v_{i\alpha_i^*}(Q(y)) - v_{i\alpha_i}(Q(y)) > c \quad \text{for all } \alpha_i \neq \alpha_i^* \in \mathcal{A}_i, \text{ and } i \in \mathcal{N}. \tag{D.20}$$

For notational convenience, we focus on player $i$ and drop the player-specific indices altogether. Let $\alpha \neq \alpha^* \in \mathcal{A}$, and define for $n \in \mathbb{N}$ the quantities $w_{\alpha,n}$ and $z_{\alpha,n}$ as

$$w_{\alpha,n} := \langle p_n, e_\alpha - e_\alpha^* \rangle, \qquad z_{\alpha,n} := \langle y_n, e_\alpha - e_\alpha^* \rangle \tag{D.21}$$

where $e_\alpha, e_\alpha^*$ are the standard basis vectors corresponding to $\alpha, \alpha^* \in \mathcal{A}$. For notational convenience, we focus on player $i$ and drop the player-specific indices altogether. Now, decomposing the IWE $\hat{v}_n$, we obtain

$$\hat{v}_n = v(x_n) + U_n + b_n \tag{D.22}$$

where $U_n := \hat{v}_n - v_i(\hat{x}_n)$ is a zero-mean noise, and $b_{i,n} := v_i(\hat{x}_n) - v_i(x_n)$.

Then, unfolding the recursion, we obtain:

$$w_{\alpha,n+1} = w_{\alpha,n} + \gamma\langle \hat{v}_n, e_\alpha - e_\alpha^* \rangle$$

$$= w_{\alpha,n} + \gamma\langle v(x_n), e_\alpha - e_\alpha^* \rangle + \gamma\langle U_n, e_\alpha - e_\alpha^* \rangle + \gamma\langle b_n, e_\alpha - e_\alpha^* \rangle$$

$$\leq w_{\alpha,n} + \gamma\langle v(x_n), e_\alpha - e_\alpha^* \rangle + \gamma\langle U_n, e_\alpha - e_\alpha^* \rangle + 2\gamma\|b_n\|_*$$

$$\leq \gamma \sum_{k=1}^{n} \langle v(x_k), e_\alpha - e_\alpha^* \rangle + \gamma \sum_{k=1}^{n} \langle U_k, e_\alpha - e_\alpha^* \rangle + 2\gamma \sum_{k=1}^{n} \|b_k\|_*$$

$$\leq \gamma \sum_{k=1}^{n} \langle v(x_k), e_\alpha - e_\alpha^* \rangle + \gamma \sum_{k=1}^{n} \langle U_k, e_\alpha - e_\alpha^* \rangle + 2\gamma B \sum_{k=1}^{n} \varepsilon_k \tag{D.23}$$

where we used that $\|b_n\|_* = \Theta(\varepsilon_n)$ for all $n \in \mathbb{N}$. Now, define the process $\{M_n\}_{n \in \mathbb{N}}$ as

$$M_n := \gamma \sum_{k=1}^{n} \langle U_k, e_\alpha - e_\alpha^* \rangle \tag{D.24}$$

which is a martingale, since $\mathbb{E}[U_n \mid \mathcal{F}_n] = 0$. Moreover, note that

$$\|U_n\|_* = \|\hat{v}_n - v(\hat{x}_n)\|_* \leq \|\hat{v}_n\|_* + \|v(\hat{x}_n)\|_* \tag{D.25}$$

i.e., $\|U_n\|_* = \Theta(1/\varepsilon_n)$. Thus, we readily obtain that $\mathbb{E}[\|U_n\|_*^q \mid \mathcal{F}_n] \leq \sigma_n^q$ for $\sigma_n = \Theta(1/\varepsilon_n)$ and all $q \in [1, \infty]$. So, by Lemma D.1 for $\gamma_n = \gamma$, $\sigma_n = \sigma$, $c$ as in Theorem 2, and $\mu \in (0,1), q > 2$ whose values will be determined next, there exists $A_q > 0$ such that:

$$\delta_n := \mathbb{P}\left( \sup_{k \leq n} |M_k| > \frac{c}{2}(\gamma n)^\mu \right) \leq A_q \frac{\gamma^{q/2+1} \sum_{k=1}^{n} \sigma_k^q}{(\gamma n)^{1+q(\mu-1/2)}}$$

$$\leq A_q \frac{\gamma^{q(1-\mu)} \sum_{k=1}^{n} \sigma_k^q}{n^{1+q(\mu-1/2)}} \tag{D.26}$$

Now, note that for $\varepsilon_n = \varepsilon/n^{\ell_\varepsilon}$, and since $\sigma_n = \Theta(1/\varepsilon_n)$, we get that there exists $M > 0$ such that

$$\sum_{k=1}^{n} \sigma_k^q \leq M \varepsilon^{-q} \sum_{k=1}^{n} k^{q\ell_\varepsilon} \tag{D.27}$$

with $\sum_{k=1}^{n} k^{q\ell_\varepsilon} = \Theta(n^{1+q\ell_\varepsilon})$. Therefore,

$$\delta_n \leq A_q' \frac{\gamma^{q(1-\mu)} \varepsilon^{-q} n^{1+q\ell_\varepsilon}}{n^{1+q(\mu-1/2)}}$$

$$\leq A_q' \frac{\gamma^{q(1-\mu)} \varepsilon^{-q}}{n^{q(\mu-1/2-\ell_\varepsilon)}} \tag{D.28}$$

Now, we need to guarantee that there exist $\mu \in (0,1), q > 2$, such that

$$\sum_{n=1}^{\infty} \delta_n < \infty \tag{D.29}$$

For this, we need to ensure that $q(\mu - 1/2 - \ell_\varepsilon) > 1$, or, equivalently,

$$\ell_\varepsilon < \mu - 1/2 - 1/q \tag{D.30}$$

which we will do later. Then, we will get for $\gamma$ small enough:

$$\mathbb{P}\left( \bigcap_{n=1}^{\infty} \left\{ \sup_{k \leq n} |M_k| \leq \frac{c}{2}(\gamma n)^\mu \right\} \right) = 1 - \mathbb{P}\left( \bigcup_{n=1}^{\infty} \left\{ \sup_{k \leq n} |M_k| > \frac{c}{2}(\gamma n)^\mu \right\} \right)$$

$$\geq 1 - \sum_{n=1}^{\infty} \delta_n$$

$$\geq 1 - \delta \tag{D.31}$$

Regarding the term $2\gamma B \sum_{k=1}^{n} \varepsilon_k$ in (D.23), we have that:

$$2\gamma B \sum_{k=1}^{n} \varepsilon_k = 2B\gamma\varepsilon \sum_{k=1}^{n} k^{-\ell_\varepsilon} \leq B'\gamma\varepsilon n^{1-\ell_\varepsilon} \tag{D.32}$$

where we used that $\sum_{k=1}^{n} k^{-\ell_\varepsilon} = \Theta(n^{1-\ell_\varepsilon})$. Thus, for

$$1 - \ell_\varepsilon < \mu \tag{D.33}$$

we have for $\varepsilon, \gamma > 0$ small enough:

$$2\gamma B \sum_{k=1}^{n} \varepsilon_k \leq B' \gamma \varepsilon n^{1-\ell_\varepsilon} \leq B' \gamma \varepsilon n^\mu \leq \frac{c}{2}(\gamma n)^\mu \tag{D.34}$$

for all $n \in \mathbb{N}$. Hence, by (D.30), (D.33) we need the following two conditions to be satisfied:

$$1 - \ell_\varepsilon < \mu \quad \text{and} \quad \ell_\varepsilon < \mu - \frac{1}{2} - \frac{1}{q} \tag{D.35}$$

for which we get that for $\ell_\varepsilon \in (0, 1/2)$, there exists always $\mu \in (3/4, 1)$ and $q$ large that satisfy (D.35). Thus, combining (D.34) and (D.31), we get by (D.23) that with probability at least $1 - \delta$:

$$w_{\alpha,n+1} \leq \sum_{k=1}^{n} \gamma_k \langle v(x_k), e_\alpha - e_\alpha^* \rangle + c(\gamma n)^\mu \quad \text{for all } n \in \mathbb{N}. \tag{D.36}$$

Thus, following similar steps as in the proof Theorem 3 after (D.14), we readily obtain that with probability at least $1 - \delta$, we have:

$$\begin{aligned} z_{\alpha,T} &\leq z_{\alpha,1} - c\gamma \sum_{k=1}^{T-1} (\gamma k - (\gamma k)^\mu) \\ &\leq z_{\alpha,1} - c\gamma^2 \frac{T(T-1)}{2} + c\gamma^{1+\mu} \int_0^T t^\mu dt \tag{D.37} \\ &\leq z_{\alpha,1} - c\gamma^2 \frac{T(T-1)}{2} + c\gamma^{1+\mu} \frac{T^{\mu+1}}{\mu+1} \end{aligned}$$

$$\tag{D.38}$$

for all $T \in \mathbb{N}$. Setting $\mu = 4/5$ and invoking Lemma A.1 for $\theta(x) = x \log x$, our claim follows. ∎

## E  Numerical experiments

In this section, we provide numerical simulations to validate and explore the performance of (FTXL). To this end, we consider two game paradigms, (i) a zero-sum game, and (ii) a congestion game.

**Zero-sum Game.**  First, we consider a 2-player zero-sum game with actions $\{\alpha_1, \alpha_2, \alpha_3\}$ and $\{\beta_1, \beta_2, \beta_3\}$, and payoff matrix

$$P = \begin{pmatrix} (2,-2) & (1,-1) & (2,-2) \\ (-2,2) & (-1,1) & (-2,2) \\ (-2,2) & (-1,1) & (-2,2) \end{pmatrix}$$

Here, the rows of $P$ correspond to the actions of player $A$ and the columns to the actions of player $B$, while the first item of each entry of $P$ corresponds to the payoff of $A$, and the second one to the payoff of $B$. Clearly, the action profile $(\alpha_1, \beta_2)$ is a strict Nash equilibrium.

**Congestion Game.**  As a second example, we consider a congestion game with $N = 100$ and 2 roads, $r_1$ and $r_2$, with costs $c_1 = 1.1$ and $c_2 = d/N$ where $d$ is the number of drivers on $r_2$. In words, $r_1$ has a fixed delay equal to 1.1, while $r_2$ has a delay proportional to the drivers using it. Note, that the strategy profile where all players are using $r_2$ is a strict Nash equilibrium.

In Fig. 1, we assess the convergence of (FTXL) with logit best responses, under realization-based and bandit feedback, and compare it to the standard (EW) with the same level of information. For each feedback mode, we conducted 100 separate trials, each with $T = 10^3$ steps, and calculated the average norm $\|x_n - x^*\|_1$ as a function of the iteration counter $n = 1, 2, ..., T$. The solid lines represent the average distance from equilibrium for each method, while the shaded areas enclose the range of $\pm 1$ standard deviation from the mean across the different trials. All the plots are displayed in logarithmic scale. For the zero-sum game, all runs were initialized with $y_1 = 0$, and we used constant step-size $\gamma = 10^{-2}$, and exploration parameter $\varepsilon = 10^{-1}$, where applicable. For the congestion game, the initial state $y_1$ for each run was drawn uniformly at random in $[-1, 1]^2$, and we used constant step-size $\gamma = 10^{-2}$, and exploration parameter $\varepsilon_n = 1/n^{1/4}$, where applicable.

The experiments have been implemented using Python 3.11.5 on a M1 MacBook Air with 16GB of RAM.

# F  Connection with other acceleration mechanisms

In this appendix, we discuss the connection between (FTXL) and the "linear coupling" method of Allen-Zhu & Orecchia [1]. Because [1] is not taking a momentum-based approach, it is difficult to accurately translate the coupling approach of [1] to our setting and provide a direct comparison between the two methods. One of the main reasons for this is that [1] is essentially using two step-sizes: the first is taken equal to the inverse Lipschitz modulus of the function being minimized and is used to take a gradient step; the second step-size sequence is much more aggressive, and it is used to generate an ancillary, exploration sequence which "scouts ahead". These two sequences are then "coupled" with a mixing coefficient which plays a role "similar" – but not equivalent – to the friction coefficient in the (HBVF) formulation of (NAG) by Su et al. [40].

The above is the best high-level description and analogy we can make between the coupling approach of [1] and the momentum-driven analysis of Su et al. [40] and/or momentum analysis in Nesterov's 2004 textbook. At a low level (and omitting certain technical details and distinctions that are not central to this discussion), the linear coupling approach of [1] applied to our setting would correspond to the update scheme:

$$x_n = Q(y_n)$$
$$w_n = \lambda_n z_n + (1 - \lambda_n)x_n$$
$$y_{n+1} = y_n + (1 - \lambda_n)\eta_n \hat{v}_n$$
$$z_{n+1} = \lambda_n z_n + (1 - \lambda_n)x_{n+1}$$

with $\hat{v}_n$ obtained by querying a first-order oracle at $w_n$ - that is, $\hat{v}_n$ is an estimate, possibly imperfect, of $v(w_n)$. The first and third lines of this update scheme are similar to the corresponding update structure of (FTXL). However, whereas (FTXL) builds momentum by the aggregation of gradient information via the momentum variables $p_n$, the linear coupling method above achieves acceleration through the coupling of the sequences $w_n, z_n$ and $x_n$, and by taking an increasing step-size sequence $\eta_n$ that grows roughly as $\Theta(n)$, and a mixing coefficient $\lambda_n$ that evolves as $\lambda_n = 1 - 1/(L\eta_n)$, where $L$ is the Lipschitz modulus of $v(\cdot)$. Beyond this comparison, we cannot provide a term-by-term correspondence between the momentum-based and coupling-based approaches, because the two methods are not equivalent (even though they give the same value convergence rates in convex minimization problems). In particular, we do not see a way of linking the parameters $\eta_n$ and $\lambda_n$ of the coupling approach to the friction and step-size parameters of the momentum approach.

In the context of convex minimization problems, the coupling-based approach of [1] is more amenable to a regret-based analysis – this is the "unification" aspect of [1] – while the momentum-based approach of Su et al. [40] facilitates a Lyapunov-based analysis. From a game-theoretic standpoint, the momentum-based approach seems to be more fruitful and easier to implement, but studying the linear coupling approach of [1] could also be very relevant.

