# OpenReview forum: "Accelerated Regularized Learning in Finite N-Person Games"
_NeurIPS.cc/2024/Conference — NeurIPS 2024 poster_

### Official Review · Reviewer_sdz4 · 2024-06-25

**Soundness:** 3
**Presentation:** 3
**Contribution:** 3
**Rating:** 6
**Confidence:** 3

**Summary:**

The paper studies the extension of Nesterov’s accelerated gradient algorithm to the solution of N-player games, named "follow the accelerated leader" (FTXL). The method is studied in both continuous and discrete time, and under various information oracles. The convergence of the algorithm is super-linear when started from an initialization close enough to the NE. Simulations are conducted which confirms the accelerated convergence numerically.

**Strengths:**

The paper is very well-written and well-motivated. The core contribution, which is the accelerated algorithm with local super-linear convergence, is clearly presented, novel, and significant to my knowledge. That the method works under different information oracles, especially bandit feedback, is also important.

Despite the simplicity, the simulations do numerically verify the accelerated rate established in theory.

**Weaknesses:**

The convergence of FTXL in all settings requires sufficiently close initialization or has to do with the neighborhood $\mathcal{U}$. The exact definition of "sufficiently close" or this neighborhood is not given in the main paper. I believe this should be added in the statement of the theorems for clarity. The authors should also discuss how realistic it is that the initial point satisfies the requirement (and/or how likely the iterates of FTXL or FTRL may run into this "good neighborhood").

It seems that no assumption on the convexity of the payoff function is made. With a general non-convex payoff function, I wonder what is the problem structure and technical apparatus that allow the authors to derive the quadratic convergence (or even convergence at all). This is currently not sufficiently discussed in the main paper. While I am happy to study the detailed analysis, I would appreciate a sketch of the proof which highlights the main techniques.

**Questions:**

Please see Weaknesses section.

**Limitations:**

Paper of theoretical nature. No negative social impact is foreseeable.

---

> ### Author Rebuttal · Authors · 2024-08-07
>
> Dear Reviewer sdz4,
>
> Thank you for your positive evaluation and constructive comments! We reply to your questions below:
>
> > The convergence of FTXL in all settings requires sufficiently close initialization or has to do with the neighborhood. The exact definition of "sufficiently close" or this neighborhood is not given in the main paper. I believe this should be added in the statement of the theorems for clarity. The authors should also discuss how realistic it is that the initial point satisfies the requirement (and/or how likely the iterates of FTXL or FTRL may run into this "good neighborhood").
>
> The determining factor for the basin of attraction of a given equilibrium $x^*$ is the minimum payoff difference $d$ at equilibrium, as per Eq. (B.1) of our paper. The neighborhood $\mathcal{U}$ in question is roughly $\mathcal{O}(d)$ in $L^1$ diameter, and it is essentially determined by the equation $u_i(x_i^*;x_{-i}) - u_i(x) \geq d/2$. We provide the relevant details in p.13, Lemma B.1 in Appendix B, and we will be happy to transfer the corresponding expressions to the main body of the paper.
>
> As for the likelihood of (FTXL) being captured by the basin of attraction of a given equilibrium, this depends on the existence or not of non-equilibrium attracting sets – such as the heteroclinic limit cycle in Jordan's matching pennies, which is universally attracting under the replicator dynamics, that is, the first-order version of (FTXL) with entropic regularization. In typical coordination / anti-coordination scenarios, the state space of the game is partitioned into basins of attraction of different strict equilibria, so convergence is almost surely guaranteed. Otherwise, if there are non-equilibrium attracting (or chain-recurrent) sets, we conjecture that (FTXL) could be captured by a spurious attractor on the boundary and fail to converge (as in Jordan's matching pennies). To the best of our knowledge, there is no theory providing a  coherent characterization of when such spurious attractors may arise in finite games; this is a very difficult problem on which very little progress has been done since the 1950's, so we cannot provide more insights here.
>
> > It seems that no assumption on the convexity of the payoff function is made. With a general non-convex payoff function, I wonder what is the problem structure and technical apparatus that allow the authors to derive the quadratic convergence (or even convergence at all). This is currently not sufficiently discussed in the main paper. While I am happy to study the detailed analysis, I would appreciate a sketch of the proof which highlights the main techniques.
>
> Please note that our paper focuses throughout on *finite* games in normal form, so the players' payoff functions are, de facto, multilinear in the players' mixed strategies – and, in particular, linear in each individual player's strategy. In the case of *continuous* games – that is, games with a finite number of players and a *continuum* of actions per player – things are dramatically different, especially if (as you suggest) there are no individual convexity assumptions made on the players' payoff / cost functions. The extension of FTXL to continuous games (possibly with a monotone structure) is a very fruitful research direction, but it would require a drastic departure from the finite game setting of the current paper, so it lies well beyond the scope of our work.
>
> Now, going back to our particular setting, the basic insight is that strict Nash equilibria have a "sharp" variational characterization, in the sense that the players' payoff vector lies in the interior of the normal cone to the point in hand (akin to Polyak's notion of sharpness for minimization problems). This allows the buildup of significant momentum accelerating the algorithm, and since strict equilibria are extreme points (all positivity constraints are saturated except one, owing to the simplex equality constraint), there is no danger of overshooting the point in question. This is the reason that the lack of friction does not hinder convergence – and, of course, it is inextricably tied to the (multi)linear structure of finite games and their strategy domains. We will of course be happy to take advantage of the extra page of the first revision opportunity to describe all this in more detail and provide a technical roadmap of the proof.
>
> ---
>
> Please let us know if you have any follow-up questions, and thank you again for your time and positive evaluation!
>
> Kind regards,
>
> The authors

---

> > ### Comment · Reviewer_sdz4 · 2024-08-08
> >
> > I thank the authors for the clarification and would like to maintain my score. I support the acceptance of the paper.

---

> > > ### Author Response · Authors · 2024-08-12
> > > **Thank you**
> > >
> > > Thank you for your time and your support!
> > >
> > > Kind regards,
> > >
> > > The authors

---

### Official Review · Reviewer_3QUL · 2024-07-07

**Soundness:** 3
**Presentation:** 3
**Contribution:** 3
**Rating:** 7
**Confidence:** 4

**Summary:**

The submission studies the convergence of finite N-person games. Combining the idea of (NAG) and (FTRL), the submission proposes a continuous-time dynamic (FTXL-D) and its discrete-time scheme (FTXL). The novelty of integrating momentum and regularization into an algorithm allows the construction of quadratic rate methods.

**Strengths:**

- A novel viewpoint that uses (HBVF)’s interpretation to link (NAG) and (FTRL-D) to form (FTXL-D).
- Improved converge rates from linear to quadratic are shown in Theorems 1 – 4. Theorems 2 – 4 cover common practical scenarios.
- Clear comparison between the proposed method and the previous methods, especially at the intuition level.
- Pinpoint the critical term to focus on the technical details (U in Theorem 3). This greatly increases the time to understand the analysis.
- Rigorous analysis is provided in the Appendix.

**Weaknesses:**

- A smoother transition toward (FTXL-D) in lines 182–186 is desired. In a game, $x$ is the response and $y$ is an aggregated payoff. For (HBVF), the interpretation is for a moving object.
  - What is the correspondence of $x$ and $y$ of a game in (HBVF)? Do we correspond $x$ (position) in (HBVF) to $y$ (payoff) in a game? If so, what corresponds to the response ($x$ in a game) in (HBVF)? Is it the control $w$ (my naive guess)?
  - These analogs, with the aid of a clearer transition, would help us to better understand the first paragraph of Section 3.2. Maybe the response x in a game and the position x in (HBVF) introduces a bit of confusion, too?
- There are two sets of notations, one for the continuous-time regime and the other for the discrete-time regime. A small table to distinguish them might help the readability.
- Although in general, the submission is easy to read, a minor aspect of improving readability could be explaining the meaning $\dot{y}$ (lines 192 and 196) and $\ddot{z}$ (line 486), which could ease the reading barrier for people who are more comfortable with discrete-time analysis.

**Questions:**

- In line 245, the friction parameter is set to zero ($r=0$) according to the discussion of Example 3. This setting, by direct substitution, would cancel out the desired effect of the momentum in (FTXL-D) in line 186 or (4) in line 197. However, in (FTXL) (line 258), the momentum appears to play an important role in the leftmost equality $y_{n+1} = y_n + \gamma p_n$. Did I misunderstand something?
- In Figure 1, why does FTXL perform poorly in the early rounds? Is it due to the constants in the bound or is there another reason?

**Limitations:**

Yes.

---

> ### Author Rebuttal · Authors · 2024-08-07
>
> Dear Reviewer 3QUL,
>
> Thank you for your strong positive evaluation and constructive comments! We reply to your questions below:
>
> > A smoother transition toward (FTXL-D) in lines 182–186 is desired. [...] What is the correspondence of $x$ and $y$ of a game in (HBVF)? [...] These analogs, with the aid of a clearer transition, would help us to better understand the first paragraph of Section 3.2.
>
> The key insight here is that, in the case of regularized learning, the algorithm's latent, state variable – that is, the variable which determines the evolution of the system in an autonomous way – is the "score variable" $y$, not the "strategy variable" $x$ (which is an ancillary variable obtained from $y$ via the regularized choice map $Q$). In particular, this means that, even though results are stated in terms of the strategy variables $x$ (which are the primary objects of interest), the true state variable of the algorithm's iterative structure is $y$, not $x$. Put differently, the $x$ in (NAG) and (HBVF) should be compared and contrasted to $y$ in (FTXL), not $x$.
>
> We made this choice of notation to be as close as possible to the work of Su, Boyd and Candès, but we understand that it may have occluded intuition. We will make sure to update the presentation here and provide more details along the above lines at the first revision opportunity.
>
>
> > There are two sets of notations, one for the continuous-time regime and the other for the discrete-time regime. A small table to distinguish them might help the readability.
>
> This is a great idea, thanks! We were (very sharply) constrained for space in the original submission, but we will be happy to take advantage of the extra page to include such a table. Thanks for the suggestion!
>
>
> > Although in general, the submission is easy to read, a minor aspect of improving readability could be explaining the meaning of $\dot y$ (lines 192 and 196) and $\ddot z$ (line 486), which could ease the reading barrier for people who are more comfortable with discrete-time analysis.
>
> Point well taken. We will include a note about this, and also make a reference to it in the proposed table of notation above.
>
>
> > In line 245, the friction parameter is set to zero ($r=0$) according to the discussion of Example 3. This setting, by direct substitution, would cancel out the desired effect of the momentum in (FTXL-D) in line 186 or (4) in line 197. However, in (FTXL) (line 258), the momentum appears to play an important role in the leftmost equality $y_{n+1} = y_n + \gamma p_n$. Did I misunderstand something?
>
> The desired effect of the momentum in (FTXL-D) – but also (NAG) and (HBVF) – is not the friction term, but the fact that the system is second-order in time, so the effects of the momentum are "baked in" the algorithm's update structure. The friction term $r\dot x$ (or $r\dot y$) has a "dampening effect" intended to mitigate the algorithm overshooting a desired state.
>
> We hope this is clearer now, please let us know if we missed or misunderstood something in your question!
>
>
> > In Figure 1, why does FTXL perform poorly in the early rounds? Is it due to the constants in the bound or is there another reason?
>
> The reason is that the algorithm is building up momentum in the initial iterations, much like uniformly accelerated motion (like pushing a crate with constant force): the algorithm starts at zero speed and is initially slow, but as it gains more and more momentum, it rapidly accelerates and gains more speed and momentum, which is ultimately responsible for the method's quadratic convergence rate. In this analogy, FTRL would correspond to uniform motion – always moving at "constant" speed, so it is faster in the initial iterations, but much slower overall since there is no momentum build-up.
>
> ---
>
> Thank you again for your strong positive evaluation and encouraging remarks – and please let us know if you have any follow-up questions!
>
> Kind regards,
>
> The authors

---

> > ### Comment · Reviewer_3QUL · 2024-08-09
> >
> > Thank you for your response. I have no further questions.

---

> > > ### Author Response · Authors · 2024-08-14
> > >
> > > Thank you again for your time, input, and positive evaluation!
> > >
> > > Kind regards,
> > >
> > > The authors

---

### Official Review · Reviewer_qtGD · 2024-07-10

**Soundness:** 4
**Presentation:** 4
**Contribution:** 4
**Rating:** 8
**Confidence:** 3

**Summary:**

The paper proposes a momentum-based follow-the-regularized-leader (FTRL) type algorithm for finding the Nash equilibrium in finite games. The paper first investigates the continuous second order ODE and a concrete game, and then devises the appropriate FTRL scheme. The convergence rate for the proposed algorithm in all three information settings of reward is quadratic (O( exp(-T^2))).

**Strengths:**

1. A rich collection of results on momentum methods for finite-games. The paper has three sets of theoretical results. First, convergence results for the second-order ODE dynamics with vanishing friction (Theorem 1) and non-vanishing friction (Theorem B.1), both under the logit best response. Second, an analysis of a momentum FTRL under full information on a concrete one-player game with vanishing (Prop C.1) and non-vanishing friction (Prop C.2). And finally, analysis momentum FTRL with three kinds of info feedback.

2. The paper discovered an important phenomenon that, in the continuous second order ODE, "the friction term hinders convergence". This claim is backed by a concrete analysis of momentum FTRL on a concrete game. In my opinion this phenomenon is interesting to the game learning community.

3. The writing is flowing and smooth.

**Weaknesses:**

See question.

**Questions:**

Minor comments
1. Are lower bounds of learning a finite game with the three kinds feedbacks known? If so, it would be great to provide a survey.
2. In line 305 the phrasing "second-order in space" is a bit confusing. In my opinion, it would be better to say "using Hessian information of relevant functions".

Typo
1. At Eq FTRL, near line 131, the second equation should be x_i,n = Q_i (y_i, n).

**Limitations:**

The paper provides a extensive discussion of momentum method for finite game solving, and does not have major limitations in my opnion.

---

> ### Author Rebuttal · Authors · 2024-08-07
>
> Dear Reviewer qtGD,
>
> Thank you for your overwhelmingly positive evaluation and constructive comments. We reply to your questions below:
>
> > Are lower bounds of learning a finite game with the three kinds feedbacks known? If so, it would be great to provide a survey.
>
> The closest lower bounds that we're aware of are algorithm-specific, e.g., as in the paper of Giannou et al. [15] where the authors argue (but do not prove) a geometric lower bound for FTRL with full information feedback, which is also achieved by FTRL with bandit feedback (so it is tight for both models). We fully share the reviewer's point of view but, regrettably, we are not aware of a more general lower-bound analysis as in the case of convex minimization.
>
> > In line 305 the phrasing "second-order in space" is a bit confusing. In my opinion, it would be better to say "using Hessian information of relevant functions".
>
> Point well taken, we will adjust the phrasing accordingly to make sure there is no confusion.
>
> > At Eq FTRL, near line 131, the second equation should be x_i,n = Q_i (y_i, n).
>
> Oh, great catch, many thanks - will fix!
>
>
> ---
>
> Please let us know if you have any follow-up questions, and thank you again for your time and overwhelmingly positive evaluation!
>
> Kind regards,
>
> The authors

---

### Official Review · Reviewer_yttq · 2024-07-20

**Soundness:** 3
**Presentation:** 3
**Contribution:** 2
**Rating:** 5
**Confidence:** 3

**Summary:**

This paper primarily focuses on introducing a Nesterov's accelerated gradient (NAG) algorithm for online learning in games.
Initially, the author shows that a continuous-time version of NAG converges to a strict Nash equilibrium at a rate that is quadratic.
This rate of convergence is notably faster than that of standard FTRL algorithms.
Following this, the paper demonstrates that the convergence rate of NAG is preserved even in a bandit feedback setting.
The experimental results confirm that the proposed algorithm converges to an equilibrium more quickly than a FTRL algorithm.

**Strengths:**

* Despite the significant importance of introducing Nesterov's accelerated gradient, a method highly esteemed in convex optimization problems, into the context of learning in games, such an approach has not been pursued until now.
* The intuition behind the proposed method is thoroughly explained. Furthermore, the proposed algorithm appears to be straightforward to implement.
* The derived convergence rates are confirmed to be tight by deriving the exact convergence rate in a simple instance.

**Weaknesses:**

My primary concern is that the theoretical results are only applicable to games with a strict Nash equilibrium.
This is somewhat problematic as even in a simple game like rock-paper-scissors, a strict Nash equilibrium does not exist.
Moreover, the convergence in games that do not have a strict equilibrium has been the subject of extensive research.
Therefore, I'm wondering how efficient NAG algorithms are in two-player zero-sum games and monotone games.

Furthermore, Theorems 1, 2, 3, and 4 hinge on the presumption that the initial strategy is in close proximity to an equilibrium.
Could you provide some insight into how close the initial point needs to be to the equilibrium?

**Questions:**

My major concerns and questions are stated in Weaknesses.
I also have the following additional questions:
* Why is the update rule of $y_{i,n+1}=y_{i,n}+\gamma p_{i,n+1}$ in (FTXL) derived? Looking at the original NAG's update rule in (NAG), it seems more natural that $y_{i,n+1}=y_{i,n}+\gamma (p_{i,n+1}-p_{i,n})$.
* Is it possible to recover FTRL algorithms by setting $r$ appropriately in FTXL?

**Limitations:**

I see no negative societal impacts that need to be addressed.

---

> ### Author Rebuttal · Authors · 2024-08-07
>
> Dear Reviewer yttq,
>
> Thank you for your input. For your convenience (and that of the committee), we reproduce and reply to your comments below one-by-one.
>
> [**Note:** all bibliography reference numbers are as in our paper]
>
> > Results are only applicable to games with a strict Nash equilibrium. [...] Therefore, I'm wondering how efficient NAG algorithms are in two-player zero-sum games and monotone games.
>
> There are several points here – related, but distinct:
> 1. *On the lack of strict equilibria.* We should begin here by stating that, by the impossibility theorem of Hart & Mas-Colell [17], there are no uncoupled learning dynamics that converge to Nash equilibrium in *all* games. In this regard, at least *some* assumption needs to be made in order to have a hope of converging to a Nash equilibrium.
>
>     Our focus on games with strict Nash equilibria is motivated by the results of [14] which state that, in the presence of uncertainty – i.e., with anything other than full, perfect information – ***only*** strict Nash equilibria can be stable and attracting with high probability. In particular, if a game does not possess a strict equilibrium, the sequence of play generated by regularized learning schemes provably fails to converge with positive probability. Thus, given that we are interested in robust convergence results that continue to hold in the presence of uncertainty, the focus on strict Nash equilibria is, in this sense, unavoidable.
>
> 2. *On zero-sum games.* A first point here is that zero-sum games may well admit strict equilibria, in which case our results apply verbatim -- for example, consider the min-max payoff matrix $[0 \quad 1; -1 \quad 0]$. In general zero-sum games, it can further be shown that FTXL identifies the *support* of the set of Nash equilibria of a zero-sum game at a quadratic, superlinear rate – though, in view of the discussion above, high probability convergence to equilibrium under uncertainty should not be expected if the support is not a singleton (e.g., as in RPS). [We did not include this more specialized result because it was beyond the scope of our submission, but we will be happy to include the above discussion in the first revision opportunity.]
>
>     Finally, even though there is an extensive literature on (usually optimistic) regularized learning methods that converge to equilibrium in zero-sum games with fully mixed Nash equilibria, it should be noted that these results typically concern learning with full, *perfect* information, as per model (10a) in our paper. These results are inherently deterministic and collapse in the presence of persistent randomness and uncertainty, see e.g., https://arxiv.org/abs/2206.06015 or https://arxiv.org/abs/2110.02134, so it is in general quite difficult to achieve convergence to Nash equilibrium without full information in zero-sum games.
>
> 3. *On monotone games.* Monotonicity is a condition that primarily concerns continuous games, that is, games with a finite number of players and *continuous* action spaces. By contrast, our paper focuses throughout on *finite* games: the mixed extension of a finite game may indeed be seen as a continuous game, but the players' (mixed) payoff functions are necessarily multilinear in their strategies. As a result, except for trivial cases (like the zero game), *only* two-player games can be monotone, and this only if the players' payoff matrices satisfy very specific conditions that ultimately make the game strategically equivalent to a zero-sum game. The extension of FTXL to (monotone) games with continuous action spaces is a very interesting one, but this is otherwise a completely different framework which lies well beyond the scope of our work.
>
> > Could you provide some insight into how close the initial point needs to be to the equilibrium?
>
> The determining factor is the minimum payoff difference $d$ at an equilibrium $x^*$ as per Eq. (B.1) of our paper. The neighborhood in question is roughly $\mathcal{O}(d)$ in $L^1$ diameter, and it is essentially determined by the equation $u_i(x_i^*;x_{-i}) - u_i(x) \geq d/2$. We provide the relevant details in p.13, Lemma B.1 in Appendix B, and we will be happy to transfer the corresponding expressions to the main body of the paper.
>
> > Why is the update rule of $y_{n+1} = y_n + \gamma p_{n+1}$ in (FTXL) derived? [...] It seems more natural that $y_{n+1} = y_n + \gamma (p_{n+1} - p_n)$
>
> The update rule in (FTXL) is derived by setting $p_n = (y_n - y_{n-1})/\gamma$ in (9), which is the direct discretization of (FTXL-D), the game-theoretic analogue of (HBVF). [Put differently, mutatis mutandis, (FTXL) is related to (FTXL-D) in the same way that (NAG) is related to (HBVF).] In our original submission, we had devoted Section 3 to provide a detailed discussion on the connection of (FTXL-D) with (NAG) and (HBVF), along with the intuition behind it, referring to the paper of Su, Boyd and Candès for the detailed relation between (HBVF) and (NAG). We will be happy to take advantage of the extra page in the revision to include the details of the link between (HBVF) and (NAG)  for completeness.
>
> > Is it possible to recover FTRL algorithms by setting $r$ appropriately in FTXL?
>
> That's an interesting question. We tried to find a way to see if it's possible to express FTRL as a limit case of FTXL, but we didn't find one.
>
> ---
>
> Please let us know if you have any follow-up questions, and thank you again for your time!
>
> Kind regards,
>
> The authors

---

> > ### Comment · Reviewer_yttq · 2024-08-13
> >
> > Thank you for your response. I have no further questions, and I will keep my score.

---

> > > ### Author Response · Authors · 2024-08-14
> > >
> > > Thank you again for your time and input.
> > >
> > > Kind regards,
> > >
> > > The authors

---

### Author Rebuttal · Authors · 2024-08-07

Dear reviewers, dear AC,

We are sincerely grateful for your time, comments, and positive evaluation!

To streamline the discussion phase, we replied to each of your questions and comments in a separate rebuttal below, and we will of course integrate all applicable points in the next revision opportunity.

Thank you again for your input and encouraging remarks, and we are looking forward to a continued constructive exchange during the discussion phase.

Kind regards,

The authors

---

### Comment · Area_Chair_y6MW · 2024-08-09
**Small question**

Hi,

Can the authors comment on whether there are any connections between their FTXL method and the "linear coupling" method of Allen-Zhu and Orecchia [1], which also aims to combine mirror descent/FTRL and acceleration? The technique I mention is not analyzed in the context of games, but I am curious of there are connections nonetheless.

Thank you!

[1] Linear Coupling: An Ultimate Unification of Gradient and Mirror Descent

---

> ### Author Response · Authors · 2024-08-12
>
> Dear AC,
>
> > Can the authors comment on whether there are any connections between their FTXL method and the "linear coupling" method of Allen-Zhu and Orecchia [1], which also aims to combine mirror descent/FTRL and acceleration? The technique I mention is not analyzed in the context of games, but I am curious of there are connections nonetheless.
>
> Excellent question, thanks for bringing it up! It's a complex issue, so we're giving two answers, one shorter and one (much) longer.
>
> ---
>
> ### **Short answer**
>
> The linear coupling approach of [1] is related to the momentum-based approach of Su, Boyd and Candès which forms the basis of (FTXL), but the two are not equivalent. In essence, they are both different approaches to acceleration – and, in turn, both different than the original "estimate sequences" approach of Nesterov – and we do not see a way of directly applying the techniques of [1] to our setting.
>
> ---
>
> ### **Long answer**
>
> Because [1] is not taking a momentum-based approach (see e.g., footnote 5 in p. 4 of the arxiv version of the paper), it is difficult to accurately translate the coupling approach of [1] to our setting and provide a direct comparison between the two methods.
>
> One of the main reasons for this is that [1] is essentially using two step-sizes: the first is taken equal to the inverse Lipschitz modulus of the function being minimized and is used to take a gradient step; the second step-size sequence is much more aggressive, and it is used to generate an ancillary, exploration sequence which "scouts ahead". These two sequences are then "coupled" with a mixing coefficient which plays a role "similar" - but *not equivalent* - to the friction coefficient in the (HBVF) formulation of (NAG) by Su, Boyd, and Candès.
>
> The above is the best high-level description and analogy we can make between the coupling approach of [1] and the momentum-driven analysis of Su, Boyd, and Candès and/or momentum analysis in Nesterov's 2004 textbook (which is itself quite different from Nesterov's original 1983 paper).
>
> At a low level (and omitting certain technical details and distinctions that are not central to this discussion), the linear coupling approach of [1] applied to our setting would correspond to the update scheme:
>
> \begin{align}
> x_t &= Q(y_t)
>     \\\\
> w_t &= \lambda_t z_t + (1-\lambda_t) x_t
>     \\\\
> y_{t+1} &= y_t + (1-\lambda_t) \eta_t \hat v_t
>     \\\\
> z_{t+1} &= \lambda_t z_t + (1-\lambda_t) x_{t+1}
> \end{align}
>
> with $\hat v_t$ obtained by querying a first-order oracle at $w_t$ – that is, $\hat v_t$ is an estimate, possibly imperfect, of $v(w_t)$.
>
> The first and third lines of this update scheme are similar to the corresponding update structure of (FTXL). However, whereas (FTXL) builds momentum by the aggregation of gradient information via the momentum variables $p_t$, the linear coupling method above achieves acceleration through the coupling of the sequences $w_t$, $z_t$ and $x_t$, and by taking an increasing step-size sequence $\eta_t$ that grows roughly as $\Theta(t)$, and a mixing coefficient $\lambda_t$ that evolves as $\lambda_t = 1 - 1/(L\eta_t)$, where $L$ is the Lipschitz modulus of $v$.
>
> Beyond this comparison, we cannot provide a term-by-term correspondence between the momentum-based and coupling-based approaches, because the two methors are not equivalent (even though they give the same value convergence rates). In particular, we do not see a way of linking the parameters $\eta_t$ and $\lambda_t$ of the coupling approach to the friction and step-size parameters of the momentum approach.
>
> In the context of convex minimization problems, the coupling-based approach of [1] is more amenable to a regret-based analysis – this is the "unification" that [1] referred to – while the momentum-based approach of Su, Boyd, and Candès facilitates a Lyapunov-based analysis. Ultimately, these are all different - though, of course, equally valid - approaches to acceleration. From a game-theoretic standpoint, the momentum-based approach seems to be more fruitful and easier to implement, but studying the linear coupling approach of [1] could also be very relevant.
>
> ---
>
> We will be happy to include a version of the above discussion at the first possible revision opportunity. In the meantime, please let us know if you have any follow-up questions on the above, and thanks again for your question and for handling our submission!
>
> Kind regards,
>
> The authors

---

### Decision · Program_Chairs · 2024-09-25

**Decision:**

Accept (poster)

**Comment:**

This paper considers the problem of learning in games. The main result is to provide a general family of algorithms called *follow the accelerated leader (FTXL)* which incorporates acceleration into the FTRL framework, and achieves a speedup over vanilla FTRL under various feedback structures.


Reviewers generally agreed that this paper makes a solid and clear contribution by integrating Nesterov-style acceleration into learning in games, and is likely to inspire further work in this direction, meriting acceptance. They also found the paper to be clear and well written.


A minor comment for the final version: The authors use the term "quadratic rate" to refer to exp(-T^2) convergence (or \sqrt{\log(1/eps)} iteration complexity to achieve precision \eps). This is different from the typical use of the term in optimization (e.g., in the context of Newton's method), where quadratic convergence typically refers to exp(-exp(T)) or \log\log(1/eps) convergence. I recommend using another term to avoid confusion here.